# Soil salinity patterns reveal changes in the water cycle of inland river basins in arid zones

Gaojia Meng[1,2,3], Guofeng Zhu[1,2,3,4,*], Yinying Jiao[1,2,4], Dongdong Qiu[1,2,4], Yuhao Wang[1,2,4], Siyu Lu[1,2,4], Rui Li[1,2,4],

Jiawei Liu[1,2,3], Longhu Chen[1,2,4], Qinqin wang[1,2,4],Enwei Huang[1,2,3], Wentong Li[1]

[1] *College of Geography and Environmental Science, Northwest Normal University, Lanzhou*

*730070, China*

[2] *Lanzhou Sub-Center, Remote Sensing Application Center, Ministry of Agriculture, Lanzhou*

*730000, China*

[3] *Shiyang River Ecological Environment Observation Station, Northwest Normal University,*

*Lanzhou 730070, China*

[4] *Key Laboratory of Resource Environment and Sustainable Development of Oasis, Lanzhou*

*730000, China*

**Corresponding author address:**

**Address:** Guofeng Zhu, College of Geography and Environment Science of Northwest Normal

University, 967, Anning East Road, Lanzhou, Gansu, China 730070

E-mail: zhugf@nwnu.edu.cn

**Abstract:** Soil salinization caused by unreasonable water resource utilization severely impacts agricultural development and ecological construction in arid inland river basins. Therefore, clarifying the water cycle mechanism of salinization in arid inland river basins is crucial for watershed ecological environment management and rational water resource utilization. Based on remote sensing and observational data, this study quantitatively analyzed soil salinization changes in the Shiyang River Basin of Northwest China's arid region from 2002-2022, and explored the impacts of water conservancy projects and agricultural irrigation on soil salinization. The results indicated that: (1) The salinization area of the Shiyang River Basin remains overall stable, but the degree of salinization is further intensifying; (2) The northern hills and oasis-desert transition zone of the Shiyang River is a region with severe salinization problems, while the central corridor plain and southern Qilian Mountain regions have lower risks of salinization; (3) Regional salinization problems are particularly prominent, caused by groundwater evaporation near reservoirs, agricultural irrigation evaporation, and downstream ecological water input evaporation. Human activities have become the decisive factor in changing the salinization pattern of inland river basins, and rational utilization and management of water resources hold tremendous potential in mitigating soil salinization.

**Keywords:** Arid zones; Soil salinization; Reservoirs; Water projects

## 1. Introduction

Land is an essential natural resource for human beings with economic, social, and ecological benefits in various production activities (Lambin and Meyfroidt, 2011; Liang et al., 2023). Soil serves as the foundation of natural ecosystems, capable of facilitating material and energy cycling within the system and maintaining interactive relationships with the biosphere, hydrosphere, and atmosphere (Seneviratne et al., 2010; Smith et al., 2015; Lehmann et al., 2020). Soil plays a critical role in promoting plant growth, regulating precipitation infiltration and distribution to coordinate watershed water cycles. Moreover, its purification capacity enables the decomposition of potential pollutants, thereby preventing water and air pollution to a certain extent

(Bünemann et al., 2018; Renshu et al., 2024). However, once soil quality declines or undergoes degradation, it can cause irreversible damage and directly impact human life (Reynolds et al., 2007; Abu Hammad and Tumeizi, 2012). Soil salinization is a critical factor in land degradation (Kramer and Mau, 2020). Its fundamental mechanism involves groundwater rising to the surface, where under high-temperature conditions, water evaporates from soil pores into the atmosphere, causing salt deposits to precipitate on the ground surface. Prolonged salt accumulation can adversely affect crop growth, ultimately leading to reduced yields and other negative consequences (Zörb et al., 2018; Folberth et al., 2016; Mao et al., 2022).

Soil salinization can be classified into primary salinization and secondary salinization based on its causes. Primary salinization is mainly influenced by natural factors, while secondary soil salinization is caused by human activities (Kaushal et al., 2005; Zhuang et al., 2021; Perri et al., 2022). In particular, improper agricultural irrigation increases the risk of elevated groundwater salinity, creating a significant challenge for fields such as hydrology and agriculture (Sharma and Minhas, 2005). Artificial water transfer projects significantly alter the connectivity between groundwater and soil water, making the trend of salt enrichment through evaporation to the surface more pronounced. Seasonal water reservoir storage can also impact soil water salinity in watersheds. Estimates suggest that global saline soil area has already exceeded 833 million hectares, with approximately 20% of agricultural land and 33% of irrigated farmland becoming salinized soils, and this situation is expected to further deteriorate (Xiao et al., 2023; Hassani et al., 2021). In arid inland river basins, the climate is extremely dry, with high soil and plant evaporation intensity and elevated groundwater levels. The degree of soil salinization is severe, with a larger affected area, particularly in Northwest China's inland regions, where salinized cultivated land accounts for nearly one-fifth of the country's total cultivated land area. Therefore, studying soil salinization in arid inland river basins is crucial for understanding watershed water cycle processes and mechanisms, and holds significant importance for agricultural irrigation and water resource management (Wichelns and Qadir, 2015).

Remote sensing technology has been widely used to assess soil salinization, and feature spectral characteristics are essential markers for identifying saline soils (Ivushkin et al., 2019). There is a significant difference in reflectance between various soil salinity levels in the visible light and near-infrared bands (Farifteh et al., 2007). Saline soils exhibit higher reflectance compared to non-saline soils and show absorption peaks in the visible light band (Zhang and Huang, 2019). There is a positive correlation between soil reflectance and soil salinization (Metternicht et al., 2003; El Harti et al., 2016; Lotfollahi et al., 2023). Saline soils show absorption peaks in the visible band, and there is a positive correlation between their soil reflectance and soil salinity. In global-scale soil salinization research, researchers have employed machine learning methods to dynamically monitor surface soil salinity over the past forty years (Hassani et al., 2020). In the context of global climate change, ML algorithms are used to predict soil salinization in the 21st century (Hassani et al., 2021). Research results indicate that salt-affected regions are primarily distributed in arid and semi-arid areas, with particular severity in Northwest China (Li et al., 2014). Due to unique climatic conditions and the influences of irrigation, drainage, and ecological water transfer, the salinization risk has further intensified in China's arid and semi-arid zones (Wang and Jia, 2012; Cañedo-Argüelles et al., 2013). The complex relationship between soil salinization and groundwater changes exacerbates regional water-salt imbalance. Irrigated agriculture carries salt through water and percolates it into groundwater layers, leading to increased groundwater salinity and subsequently triggering soil salinization in irrigation areas (Foster et al., 2018). Moreover, with the continuous expansion of agricultural land, the excessive development of land resources has produced profound and persistent impacts on soil salinization (Wang and Li, 2013).

The Shiyang River Basin is located in the arid northwest of China, representing a typical inland river system, with its soil salinization problem progressively worsening due to water infrastructure and irrigation activities. Therefore, assessing the distribution of soil salinization in this basin is crucial for understanding how natural and human activities impact soil salinization in arid areas. In this study, we aim to

address the following questions: (1) Quantitatively analyze the degree of salinization in the Shiyang River Basin and reveal its spatial and temporal distribution characteristics; (2)Analyze the impacts of water cycle changes on salinization. The study's results will help clarify the impact of the water cycle on soil salinization in the inland river basin and provide a scientific basis for agricultural development, ecological construction, and water resource use planning in the arid zone.

## 2. Materials and Methods

### 2.1 The Background Conditions of the Study Area

The Shiyang River Basin is located in northwestern China, at the eastern end of the Hexi Corridor. It consists of eight major tributaries: the Dajing River, the Gulang River, the Huangyang River, the Zamu River, the Jinta River, and the Xiyang River (Fig. 1). Lakes and wetlands in the whole region mainly exist in reservoirs, with 15 reservoirs built with a more than 1 million cubic meters capacity. Water storage in reservoirs helps to adjust the distribution of river water. The study area is located in the BWK climate zone under the Köppen-Geiger climate classification, which is a cold arid desert climate (Beck et al., 2018; Beck et al., 2023). It features strong solar radiation, intense evaporation, significant diurnal temperature variation, and an annual average temperature below 8°C. Precipitation is sparse and primarily influenced by westerlies and monsoons. Mountain areas receive more precipitation than plains, with higher precipitation during summer and autumn, and significantly less during winter and spring. The terrain slopes from southwest to northeast and is divided into three units. The bedrock of the southern Qilian Mountains consists of metamorphosed sandstones and volcanic rocks, with soil textures predominantly coarse and medium, including Cryosols, Leptosols, and Phaeozems (WRB, 2022). The land is primarily forest and grassland, with annual precipitation of 300-600mm, evaporation rates of 700-1200mm, and the groundwater level is 50-200 meters below the surface. The central corridor plain features bedrock composed of schist and slate, with soil textures predominantly medium and fine, including Gypsisols, Calcisols, and Solonchaks. The land use is primarily agricultural, with annual precipitation of 150-300mm, evaporation rates of 1300-2000mm, and the groundwater level is 15-50 meters below

the surface. The bedrock of the northern hills and oasis-desert transition zone is

predominantly igneous rock, with soil textures mainly coarse, including Arenosols,

Leptosols, and Solonchaks. The landscape is barren, with annual precipitation below

150mm, evaporation rates of 2000-3000mm, and the groundwater level is 10-30

meters below the surface. The three geomorphological units show distinct differences,

with increasing aridity from south to north.

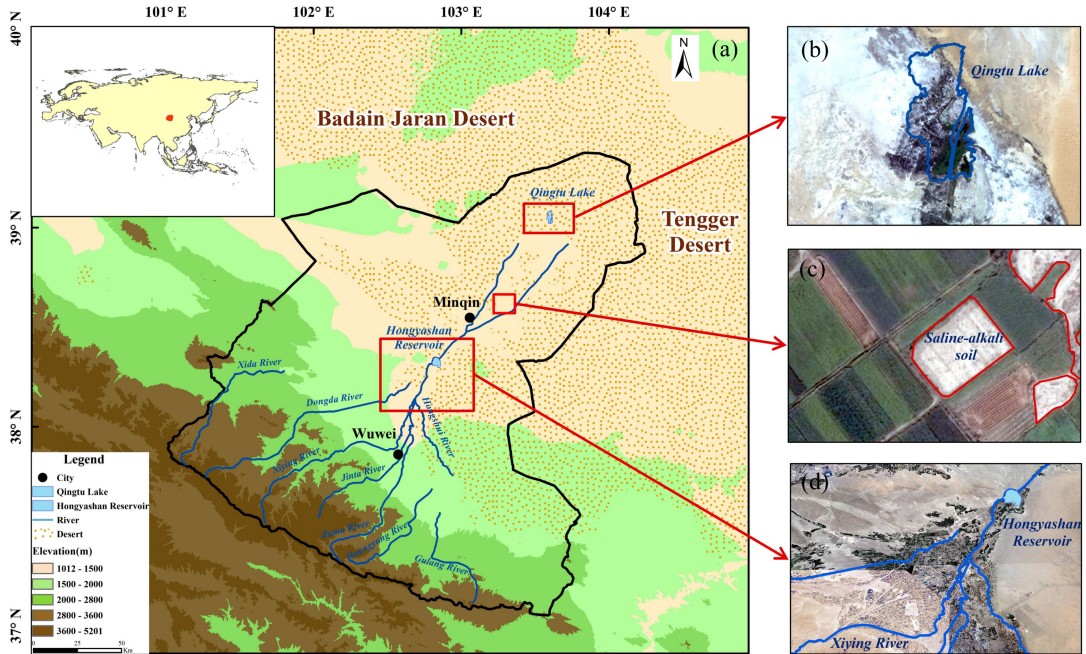

**Figure 1.** Overview map of the study area (a:Location distribution map of the Shiyang River Basin;

b: Qingtu Lake (from USGS); c: Saline soils in agricultural land (from Google Maps); d:

Distribution of water systems in the Shiyang River Basin (from USGS))

**2.2 Data Sources**

2.2.1 Remote sensing data

The Landsat series from the United States is jointly managed by the National

Aeronautics and Space Administration (NASA) and the United States Geological

Survey (USGS). It is a series of Earth observation satellite systems used by the U.S.

for monitoring Earth's resources and environment. Landsat is mainly used to

investigate ocean resources and groundwater resources and to assist in regulating the

rational use of water resources. Landsat data is available from the Earth Explorer

service, which provides surface reflectance every 16 days with a spatial resolution of

30 meters (Table 1). This article uses satellite data from Landsat-5, Landsat-7, Landsat-8, and Landsat-9. Landsat-5 was launched in March 1984, carrying the Multispectral Scanner (MSS) and Thematic Mapper (TM), and provided nearly 29 years of Earth imaging data. Landsat-7 was launched in April 1999, carrying the Enhanced Thematic Mapper Plus (ETM+) and the SLC sensor. Since June 2003, this sensor has collected and transmitted data with gaps caused by the failure of the scan line corrector (SLC), providing better radiometric and geometric data. Landsat-8, launched in February 2013, carries the Operational Land Imager (OLI) and the Thermal Infrared Sensor (TIRS), ensuring the continuity of land data reception and availability. Its data is consistent with existing standard Landsat data products. The OLI sensor is used to capture remote sensing images in the visible, near-infrared, and shortwave infrared spectral ranges and is designed with a push-broom configuration, providing better level and stability and resulting in higher quality images. The OLI-2 sensor on Landsat-9 has a higher radiometric resolution, enabling finer detection in areas such as water bodies and dense forests.

2.2.2 Land use Data

This study obtained a land cover dataset for the Shiyang River Basin at a 30-meter resolution, covering the time period from 2002 to 2022 (Table 1).

2.2.3 Digital Elevation Model

The Digital Elevation Model (DEM) data is derived from the ASTER GDEM dataset, jointly developed by Japan's METI and the U.S. NASA. The resolution is 30m, and slope data is calculated from the DEM data.

**Table 1.**List of data products used in the study

| Products | Temporal resolution | Spatial resolution | Temporal coverage | Data Source |
|---|---|---|---|---|
| Landsat-5 | 16d | 30m | 1984-2013 | https://earthxplorer.usgs.gov |
| Landsat-7 | 16d | 30m | 1999-present | https://earthxplorer.usgs.gov |
| Landsat-8 | 16d | 30m | 2013-present | https://earthxplorer.usgs.gov |
| Landsat-9 | 16d | 30m | 2021-present | https://earthxplorer.usgs.gov |
| Land use | Annual | 30m | 1985-2022 | https://zenodo.org/records/8176941 |
| ASTER GDEM | / | 30m | 2000-2019 | http://reverb.echo.nasa.gov/reverb/ |

**2.3 Data preparation and processing**

This study selected 2002, 2007, 2012, 2017, and 2022 as research time nodes,

with four satellite remote sensing images captured for each period. To ensure the

quality of remote sensing images and salinization identification accuracy, we

prioritized obtaining high-quality images with cloud cover less than 1% during

summer, which is more conducive to identifying soil salinization levels (Allbed and

Kumar, 2013). During the remote sensing image processing stage, we used ENVI 5.3

(sourced from https://www.l3harrisgeospatial.com/Software-Technology/ENVI) to

preprocess the images, including radiometric calibration, atmospheric correction,

image fusion, image mosaic, and image clipping. Based on the natural properties of

the study area's soil, auxiliary data, and field survey information, we selected

interpretation marker points for mild, moderate, and severe saline lands, as well as

other land types using high-resolution Google Maps imagery. For saline land

extraction, we adjusted the optimal band combination for satellite remote sensing

images (Khan et al., 2005; Jia et al., 2024), and combined the Normalized Difference

Salinity Index (NDSI), slope data, and texture features. We employed support vector

machine algorithms for supervised classification of the study area to precisely identify

the spatial distribution of saline lands. The formula is as follows:

$$\min_{w,\, b,\, \xi_i} \left( \frac{1}{2} \|w\|^2 + C \sum_{i=1}^{n} \xi_i \right) \tag{1}$$

$$y_i\left(w \cdot x_i\right) \geq 1 - \xi_i,\, \xi_i \geq 0,\, i = 1,\ldots,n \tag{2}$$

In the formula, $w$ represents the weight vector, which defines the direction of the

hyperplane; $b$ is the bias term, defining the offset of the hyperplane; $\xi_i$ is the slack

variable, which increases the robustness of the model; $C$ is the regularization parameter,

balancing the model complexity and training error; $y_i$ is the label of data point $i$,

commonly used to define a hyperplane.

Finally, the accuracy of the supervised classification results is evaluated using

the confusion matrix method, including overall classification accuracy, Kappa

coefficient, etc. The data processing flow is shown in Figure 2.

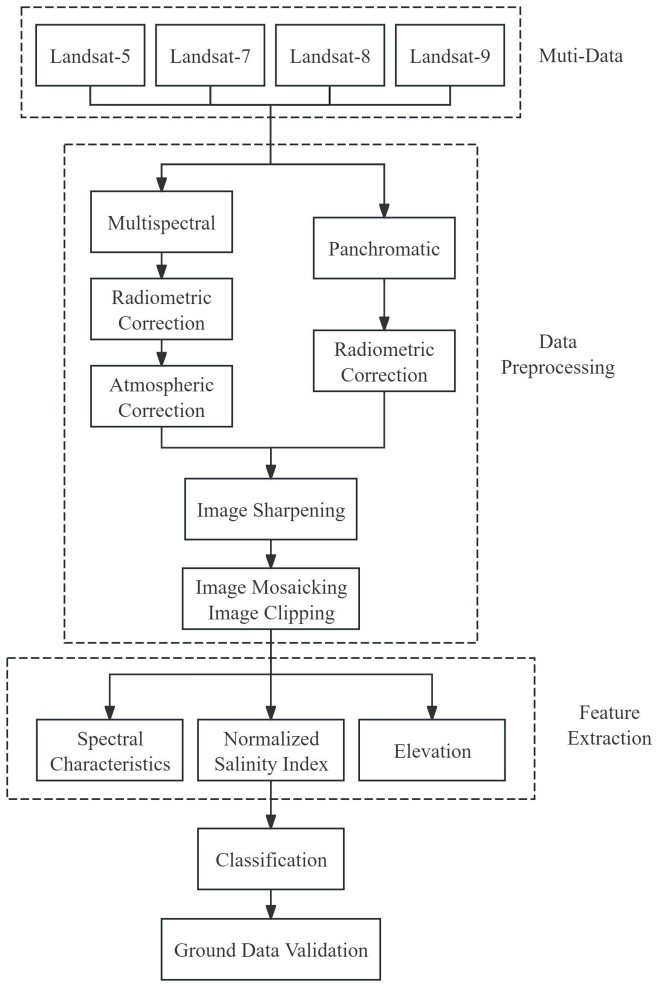

**Figure 2.**Flow chart of data processing

**2.4 Remote sensing data Validation**

2.4.1 Validation using soil salinity observation data

We selected 11 soil sampling sites within the Shiyang River Basin during the

period of 2019-2024. Through systematic field investigation and sampling, we

measured the electrical conductivity and pH values of soil samples in the laboratory.

The spatial distribution of sampling points and dataset validation points is illustrated

in Figure 3a, with the measured electrical conductivity and pH values meticulously

recorded in Table 2. The measured electrical conductivity and pH values serve as

ground truth data, forming a mutual verification relationship with remote sensing

interpretation data. From the data, it can be observed that the electrical conductivity is

high (13.65-15.67 dS/m) in the wetland areas around Qingtu Lake and Hongyashan

Reservoir in the northern part of the Shiyang River Basin, while it is generally low

(1.20-2.02 dS/m) in the upstream areas at higher elevations. This spatial variation pattern is highly consistent with the salt characteristic features identified from the corresponding remote sensing image patches. The pH value ranges from 8.30 to 8.51, and its variation trend is consistent with the salt content variation.

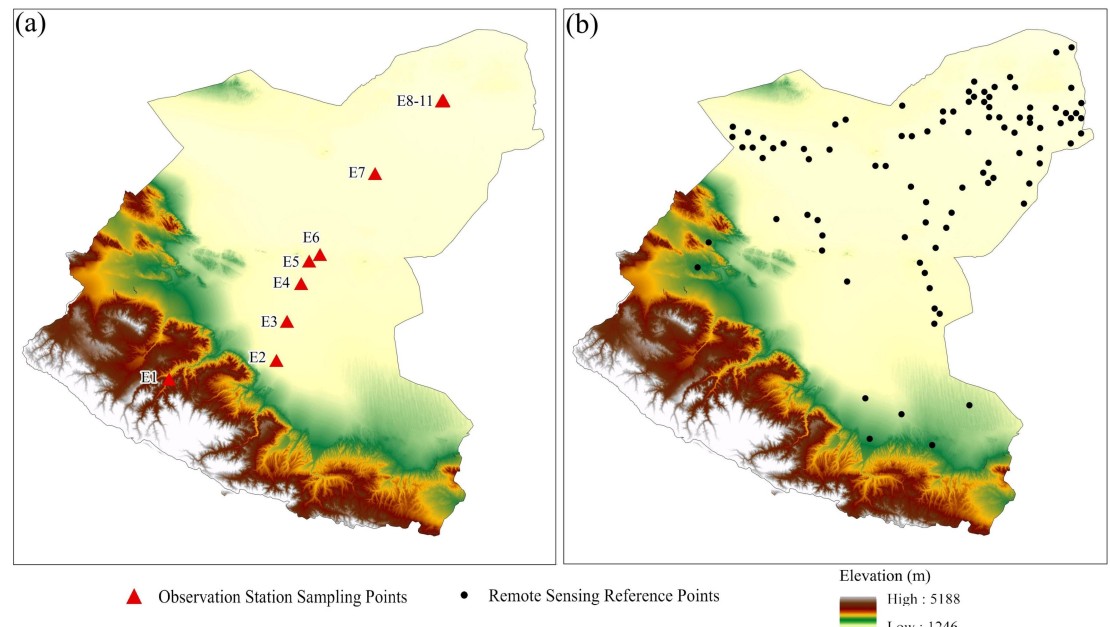

Figure 3. Distribution of Validation Points in the Shiyang River Basin

Table 2. Soil Sampling Points in the Shiyang River Basin

| Number | Sampling point | Electrical Conductivity(dS/m) | PH |
|---|---|---|---|
| E1 | Huajian Township | 1.20 | 8.41 |
| E2 | Wuwei Midstream | 1.45 | 8.46 |
| E3 | Wuwei Basin | 2.02 | 8.44 |
| E4 | Dongtan Wetland | 13.82 | 8.36 |
| E5 | Hongyashan Reservoir Inlet | 9.50 | 8.37 |
| E6 | Hongyashan Reservoir Outlet | 14.43 | 8.30 |
| E7 | Datan Township | 1.46 | 8.39 |
| E8 | 50m West of Qingtu Lake | 13.91 | 8.31 |
| E9 | 50m East of Qingtu Lake | 15.54 | 8.36 |
| E10 | 100m West of Qingtu Lake | 13.65 | 8.44 |
| E11 | 100m East of Qingtu Lake | 15.67 | 8.51 |

2.4.2 Cross-validation of remote sensing data

We used the HWSD 2.0 data released by the Food and Agriculture Organization (FAO) and extracted the relevant data for our research area. Based on the salt-alkali patches currently identified by remote sensing, we selected 92 verification points and extracted their corresponding soil information (Fig. 3b). After verification, we found

that among these 92 verification points, 75 points were consistent with the interpretation results, resulting in an overall accuracy of 81.52%.

2.4.3 Control Experiment Validation for River Basin

Field validation of remote sensing data authenticity conducted at the experimental field in Gulang County of Shiyang River Observation Station, Northwest Normal University in 2024(Fig. 4), with specific steps including: delineating experimental plots - collecting soil samples - laboratory analysis to determine actual soil salinity - conducting remote sensing monitoring of vegetation and salinity throughout the entire growing season - soil sampling throughout the entire growing season - comparative analysis of remote sensing monitoring results and experimental data for assessing their discrepancies and consistency. In the soil salinization area of Gulang County in the Shiyang River Basin (Fig. 3), the electrical conductivity measurement values were compared with the Salinization Index (Table 3). The results demonstrated that the electrical conductivity changes in saline soil samples were consistent with remote sensing monitoring results, with a high correlation coefficient, indicating a significant correlation between the two, thus the reliability of the remote sensing monitoring technology used.

Table 3. Soil Sample Electrical Conductivity and SI Comparison (EC units:dS/m)

| County | Month | Mild Soil Salinization | | Moderate Soil Salinization | | Severe Soil Salinization | |
|---|---|---|---|---|---|---|---|
| | | EC | SI | EC | SI | EC | SI |
| Gulang | March | 3.63 | 4.37 | 11.46 | 4.23 | 14.67 | 4.45 |
| | April | 4.21 | 4.17 | 11.00 | 4.14 | 13.81 | 4.28 |
| | May | 4.06 | 4.10 | 10.70 | 4.07 | 13.67 | 4.20 |
| | June | 2.43 | 3.01 | 10.37 | 5.67 | 13.49 | 3.68 |
| | July | 3.06 | 3.30 | 10.49 | 4.53 | 13.20 | 3.90 |
| | August | 2.55 | 3.20 | 10.61 | 3.30 | 12.87 | 3.34 |

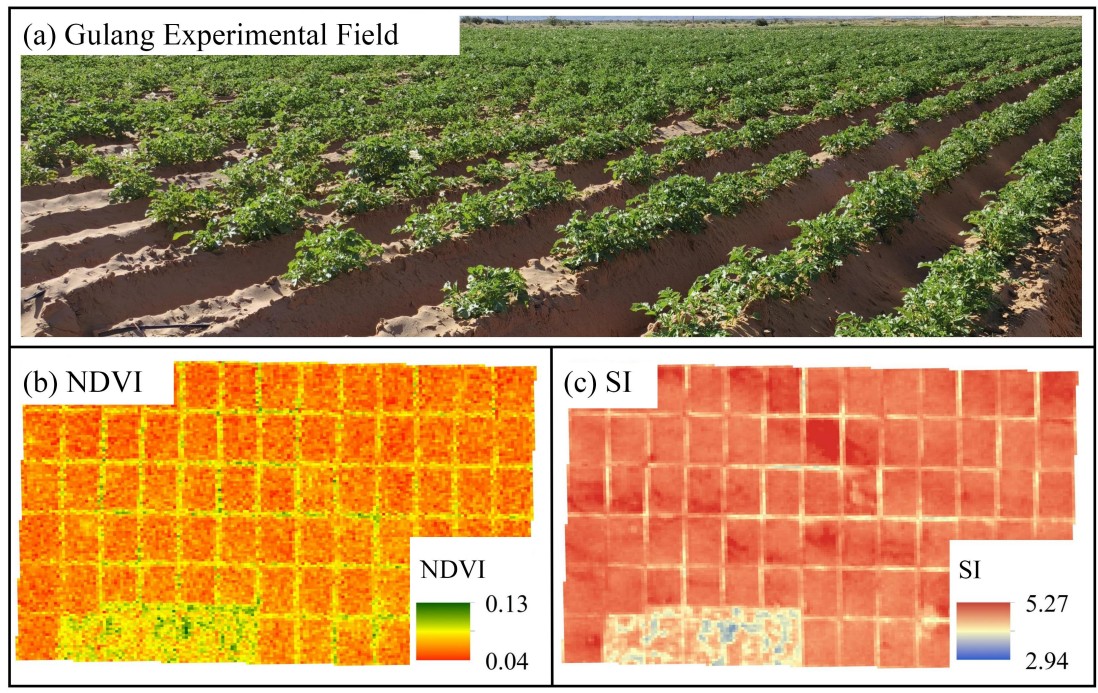

Figure 4. Experimental Field

## 3. Results

### 3.1 Spatial distribution of soil salinization

This study conducted remote sensing inversion of soil salinization in the Shiyang River Basin from 2002 to 2022 (Fig. 5). The results showed that the salinization of the basin gradually increased from upstream to downstream, especially in the downstream of the basin near Qingtu Lake, where the salinization of the soil was the most serious. From the perspective of natural landform division, the salt-accumulating areas of the Shiyang River Basin are widely distributed across the central corridor plains, northern hills and oasis-desert transition zone. In the central corridor plains, soil salinization is mainly characterized by mild and moderate salinization. Moderate saline soils are primarily concentrated in the oasis farmland irrigation areas on both sides of the river, with a few plots transforming into severe saline soils. In the northern hills and oasis-desert transition zone, soil salinization is mainly characterized by moderate and severe salinization, with the area and extent far exceeding those of the central corridor plains.

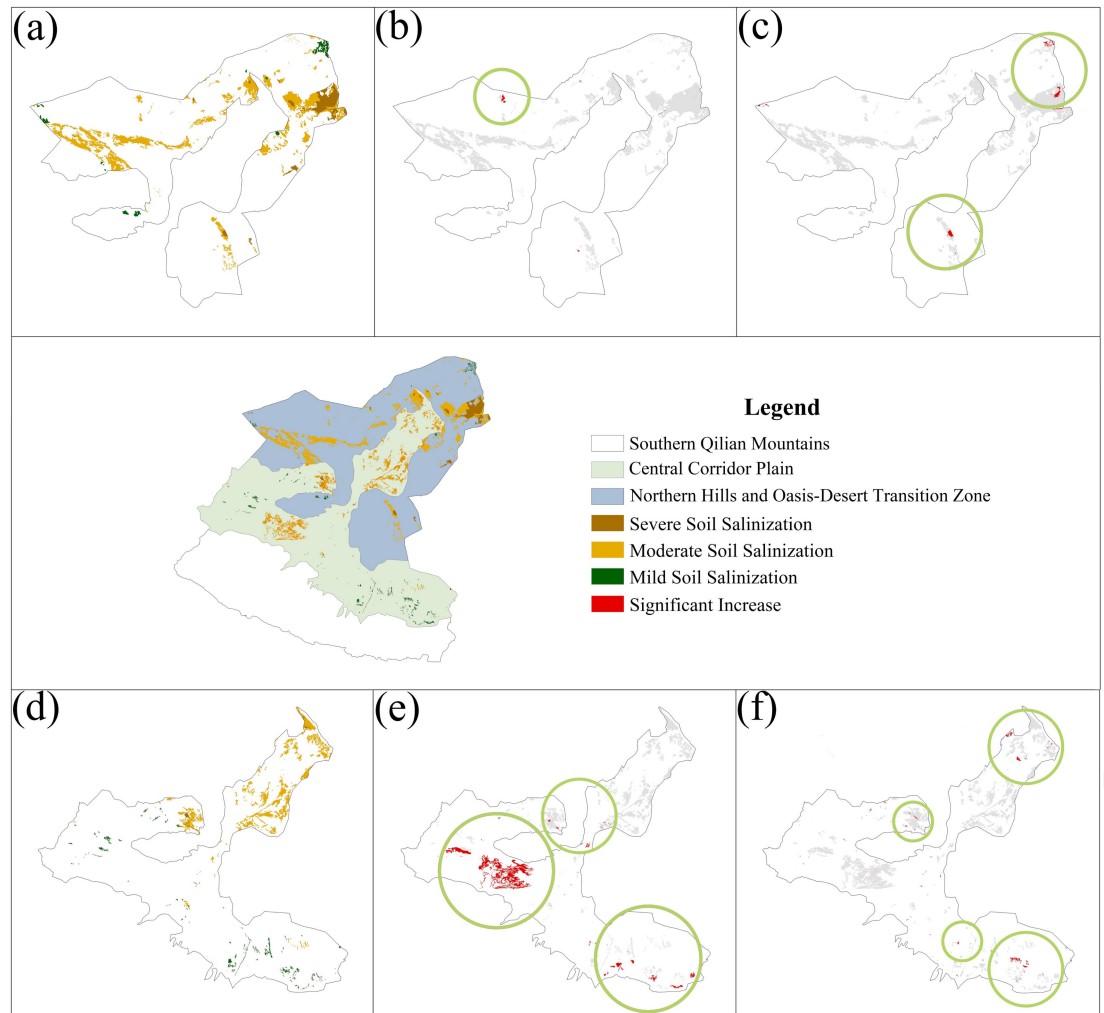

**Figure 5.** Spatial Distribution Map of Salinization in the Shiyang River Basin (a: Distribution of soil salinization in the northern hills and oasis-desert transition zone in 2002; b-c: Expansion areas in soil salinization in the northern hills and oasis-desert transition zone in 2012 and 2022; d: Distribution of soil salinization in the central corridor plain in 2002; e-f: Expansion areas in soil salinization in the central corridor plain in 2012 and 2022)

**3.2 Temporal changes in soil salinisation**

The study area is divided into three parts based on natural landform units. The Southern Qilian mountain area did not exhibit soil salinization, so the study focused on analyzing the temporal changes in soil salinization area in the central corridor plains and the northern hills and oasis-desert transition zone (Fig. 6). From 2002-2022, the total soil salinization area in the Shiyang River Basin showed an upward trend, with significant changes in its spatial distribution pattern. The salinization area decreased from 2002-2007, increased from 2007-2012, and then decreased again from

2017-2022. The central corridor plain of the Shiyang River Basin experienced a

significant increase in salinization area, with moderate salinization showing the most

notable increase, while mild and severe salinization zones remained relatively stable.

Compared to 2002, the soil salinization area in 2022 increased by over 18%. The area

of mild salinization reached its peak in 2017 and slightly decreased by 2022.

Moderate salinization area showed the most pronounced upward trend, while severe

salinization area experienced minimal changes. In the northern hills and oasis-desert

transition zone, the soil salinization area overall demonstrated a downward trend, with

an average annual decline rate of 38.19 km². Mild salinization area slightly increased.

Moderate salinization area continuously decreased, with the most significant

reduction occurring between 2002-2007. Severe salinization area slowly decreased

from 2002-2017 but increased from 2017-2022. Overall, the area of soil salinization

increased in the central corridor plain, while it remained relatively stable in the

northern hills and oasis-desert transition zone. The total soil salinization across the

entire basin showed a slight overall increase.

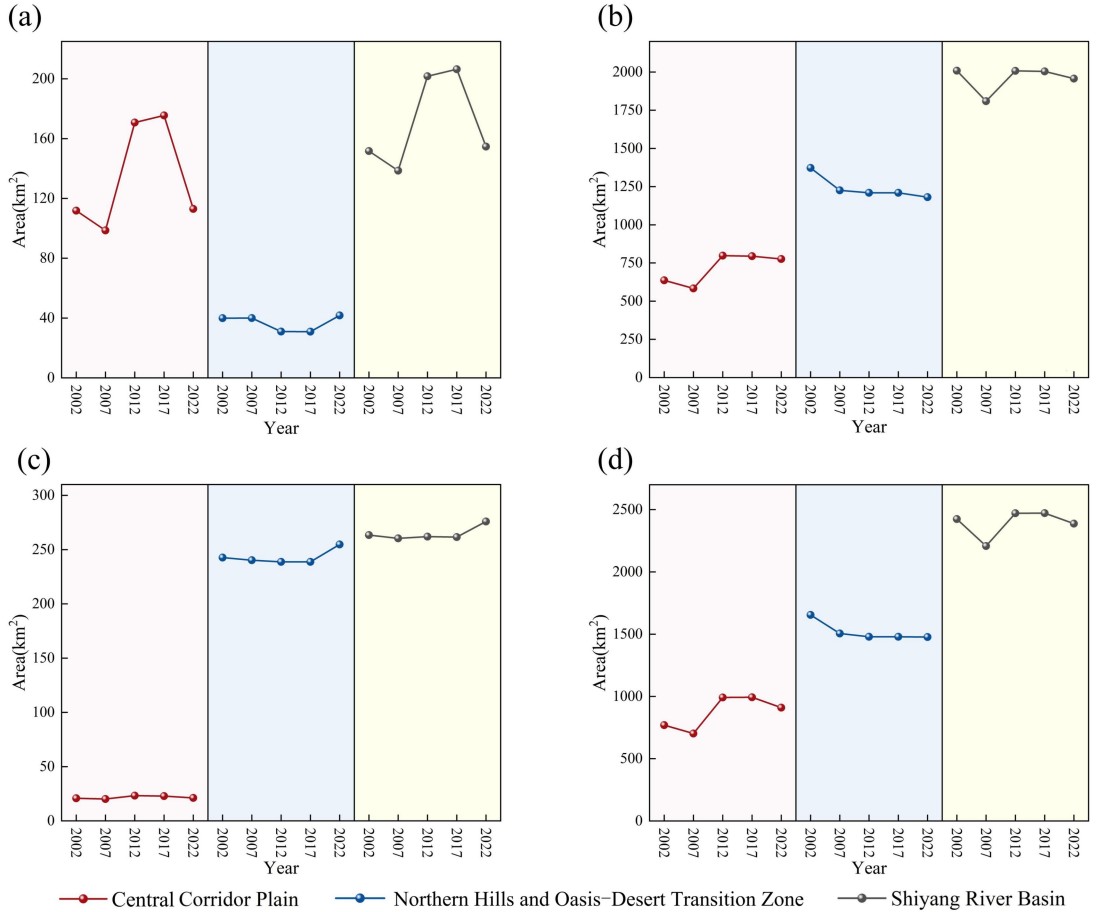

**Figure 6.** Changes in Soil Salinization Area in the Shiyang River Basin (a~d: Changes in areas of mild, moderate, severe, and total soil salinization in the Shiyang River Basin)

**4 Discussion**

**4.1 Soil salinization and basin water conservancy project**

The Shiyang River Basin is a typical inland river basin whose geographical and climatic characteristics provide a unique background for studying salinization phenomena, making it an ideal area for researching salinization processes (Ji et al., 2006; Zhu et al., 2022). Long-term monitoring through multiple observation points in the basin has revealed that the salinization issue shows a tendency to worsen. As water diversion projects advance and water transfer volumes increase, agricultural irrigation water will inevitably increase significantly (Rui and Hang, 2023). The input of external water will necessarily disrupt the balanced state between regional soil, vegetation, and climate, thus requiring careful attention to salinization issues arising from agricultural irrigation (Abbas et al., 2013; Thorslund et al., 2021). In the long

term, secondary salinization is poised to become the primary potential obstacle to sustainable inter-basin water transfer (Karimzadeh et al., 2024). Its negative effects are reflected in two aspects: the altered evaporation process following water irrigation and the groundwater level rise caused by external water sources (Duan et al., 2022). The connectivity between groundwater and soil moisture increases, making the trend of salt concentration through evaporation to the surface more pronounced. In arid regions with low rainfall and high evaporation, this will lead to salt accumulation in the soil surface from dissolved salts (Aboelsound et al., 2023). In the upstream of the Shiyang River Basin, natural water sources such as precipitation and snowmelt are introduced into irrigation districts (Zhu et al., 2022); the middle reaches improve water resource supply by constructing reservoirs and channels (Sang et al., 2023); the downstream primarily relies on upstream Shiyang River water and ecological water transfers (Qiu et al., 2023). In the basin's ecological water transfer, water diversion from the Hongyashan Reservoir to Qingthu Lake is a crucial measure for adjusting irrigation water patterns (Fig.7). The Jingdian Phase II water diversion project started in 2001, continuously introducing Yellow River water to downstream areas. These water conservancy projects have effectively alleviated water resource constraints in the basin's downstream to some extent, but may lead to further accumulation of soil salinity.

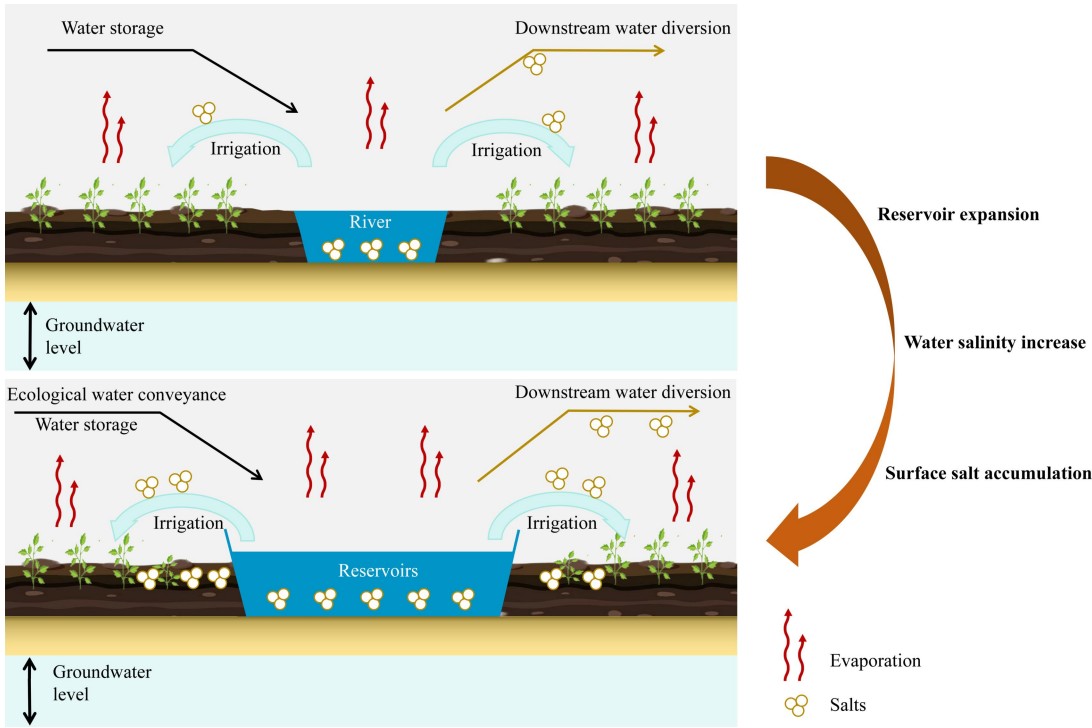

**Figure 7.** The process of salinization caused by reservoirs

Reservoir construction is an important cause of soil salinization in surrounding agricultural lands (Wu et al., 2019). The Hongyashan Reservoir is located in the desert downstream of the basin, with its western side built against the Hongya mountains and other dams artificially constructed. The reservoir aims to improve downstream ecological water scarcity, but as the reservoir continues to expand further, the downstream water storage gradually declines. From 2002-2022, the soil salinization in areas surrounding the Hongyashan Reservoir gradually intensified (Fig.8). In 2002, the reservoir area was relatively small, with severe and moderate soil salinization appearing in its western part, while some moderate soil salinization also emerged in the southeastern part. By 2007, water storage in the western part increased, and the salinization zone in the southeast shifted southward. In 2012, soil salinization in the southern part of the reservoir became severely acute, turning into severe salinization, and severe salinization also appeared in the southwestern. From 2012-2022, the salinized area around the reservoir further expanded. Additionally, measurements of groundwater electrical conductivity in the Hongyashan Reservoir from 2017-2019 revealed that the EC values consistently remained above 500μs/cm, with a slight upward trend in recent years, increasing by 14.119μs/cm from 2017-2023. The

increased groundwater salinity around the reservoir, combined with the extremely arid climate and low rainfall, intensifies surface water evaporation, leading to salt accumulation in the surface soil and gradually intensifying soil salinization (Yang et al., 2020; Yin et al., 2022).

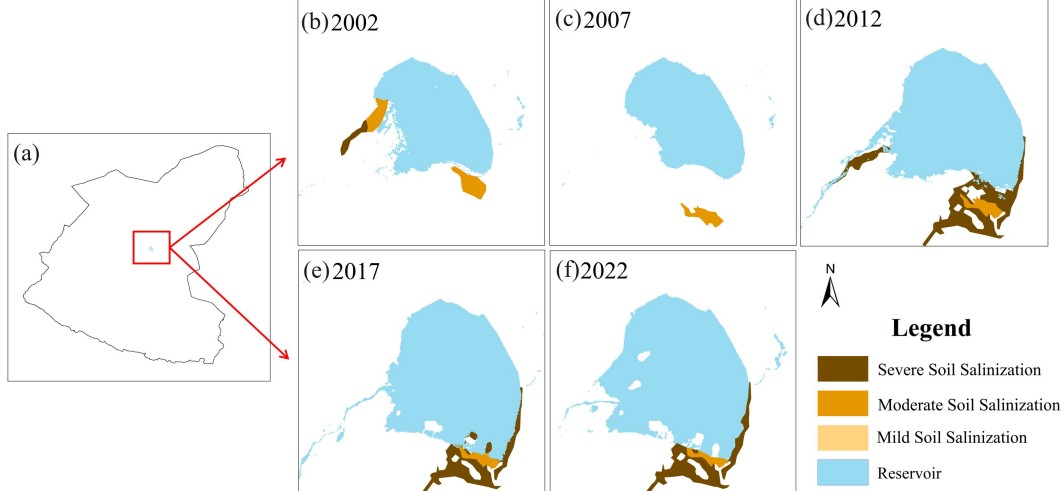

**Figure 8.** Changes in Water Body Area and Surrounding Soil Salinization of Hongyashan Reservoir

### 4.2 Soil salinization and irrigation

Developing irrigated agriculture is necessary to meet the continuously growing global food demand (Jägermeyr et al., 2017). In agricultural production, irrigation is a critical cause of salinization. Clarifying the relationship between salinization and irrigation and providing potential solutions is crucial. From 2002 to 2007, the basin's irrigation area increased from 5,131.35 km² to 5,381.58 km², showing a significant upward trend, while the salinization area showed a contrary downward trend. From 2007 to 2012, the irrigation area decreased, but the basin's salinization area increased. From 2012 to 2017, the basin's irrigation area notably decreased by about 100 km², while the salinization area remained relatively unchanged. From 2017 to 2022, the irrigation area rebounded significantly, exceeding 5,300 km², simultaneously with a substantial decline in salinization area. In the short term, an expansion of irrigation area is usually accompanied by a reduction in salinization area, while a decrease in irrigation area corresponds to an increase in salinization area. This phenomenon

indicates that moderate irrigation can temporarily reduce salt concentration in the soil surface layer through leaching, alleviating soil salinization. However, from a long-term perspective, salinized lands in the Shiyang River Basin are primarily distributed within irrigation areas, revealing another causal relationship between irrigation and salinization. Without scientific irrigation management, prolonged over-irrigation coupled with inadequate drainage systems leads to groundwater level rise, causing deep-layer salts to move upward, while dissolved salts in irrigation water accumulate in the soil through water evaporation (Minhas et al., 2020). This cumulative effect ultimately exacerbates soil salinization problems. From 2002-2022, the conversion rate of fallow land to saline land was 7.11%, significantly higher than grasslands (5.68%) and cultivated lands (2.92%). This difference highlights the critical role of continuous irrigation in suppressing soil salinization, but also reveals that without effective irrigation and salt elimination mechanisms, the region's agricultural ecosystem faces extremely high salinization risks.

In the basin's irrigation area, irrigation modes significantly impact soil salinization in both agricultural and non-agricultural regions (Fig. 9). Irrigation increases soil moisture replenishment, triggering more complex mechanisms in the vegetation root zone. When irrigation water exceeds plant root absorption capacity and evaporation rates, excess water seeps into the groundwater system, causing groundwater level rise. During irrigation process, soil moisture and salts are leached into soil layers and concentrate towards the surface through plant root water absorption (Raats, 1974; Gao et al., 2024). Soil salinization leads to vegetation degradation or affects vegetation diversity (Perri et al., 2020; Jiao et al., 2021). Lands outside the irrigation area also face salinization risks. Although leaching soil salts within the irrigation area can somewhat improve salinization conditions, the resulting high-salinity water can seep into areas outside the irrigation zone. As water evaporates, salts accumulate in the soil surface, exacerbating soil salinization problems (Singh, 2022). Additionally, due to the unique geographical environment of the Minqin oasis region, livestock farming is an important industry. However, excessive grazing leads to soil compaction, reducing soil porosity and decreasing soil permeability. Water

struggles to penetrate deeply into the soil, further exacerbating the soil salinization.

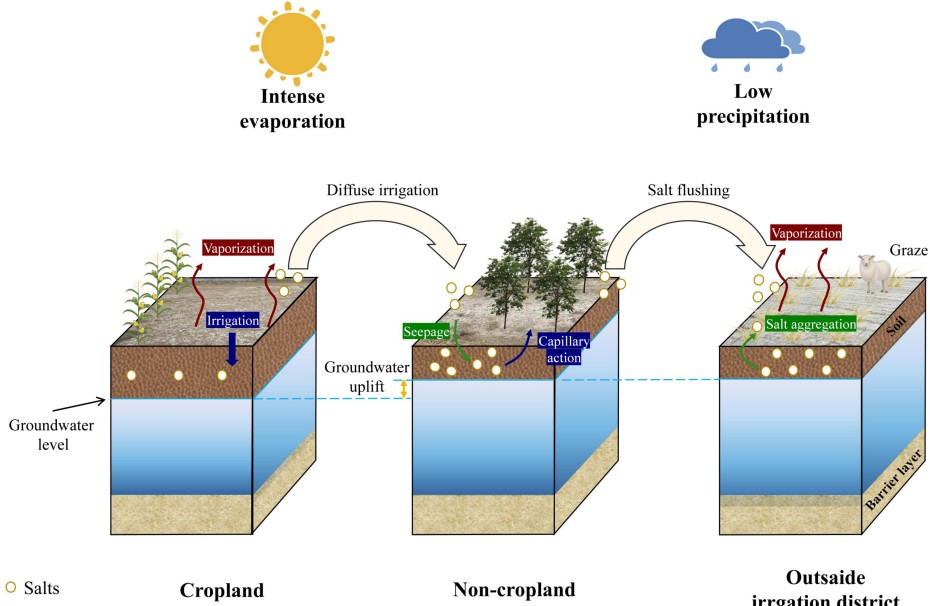

**Figure 9.** The process of salinization caused by agricultural irrigation

Surface water and groundwater irrigation are the primary irrigation methods in the Shiyang River Basin, significantly impacting soil salinization in both agricultural and non-agricultural areas (Fig.9). The Shiyang River Basin comprises 27 irrigation districts (Fig.10), with seriously salinized districts concentrated in the middle and lower reaches, while non-salinized districts are located in the upstream region. Irrigation districts with severe soil salinization include Hongyashan, Changning, Donghe River, Nanhu, Donghe, Xiyinghe River, and Qinghe districts. Among these, the Donghe River district experienced particularly severe soil salinization, with a significant increase in salinized area during 2007-2012. Districts with lighter salinization include Gulang River, Wujiaojing, Huangyang River, Yinhuang, Qiduntai, Jingdian, Dajing River, Qingyuan Well, Zamu River, Jinta River, Jinyang Well Source, Yongchang, Xihe, Siba, and Jinchuan districts, with relatively small salinized areas. The Gulang River and Wujiaojing districts showed a continuous increase in salinized area from 2002 to 2017, but experienced a reduction from 2017 to 2022, while other districts saw minimal changes in salinization area. Overall, irrigation is the main factor influencing the gradual increase in soil salinization from upstream to downstream in the Shiyang River Basin, highlighting the profound impact of human agricultural activities on salinization in the basin.

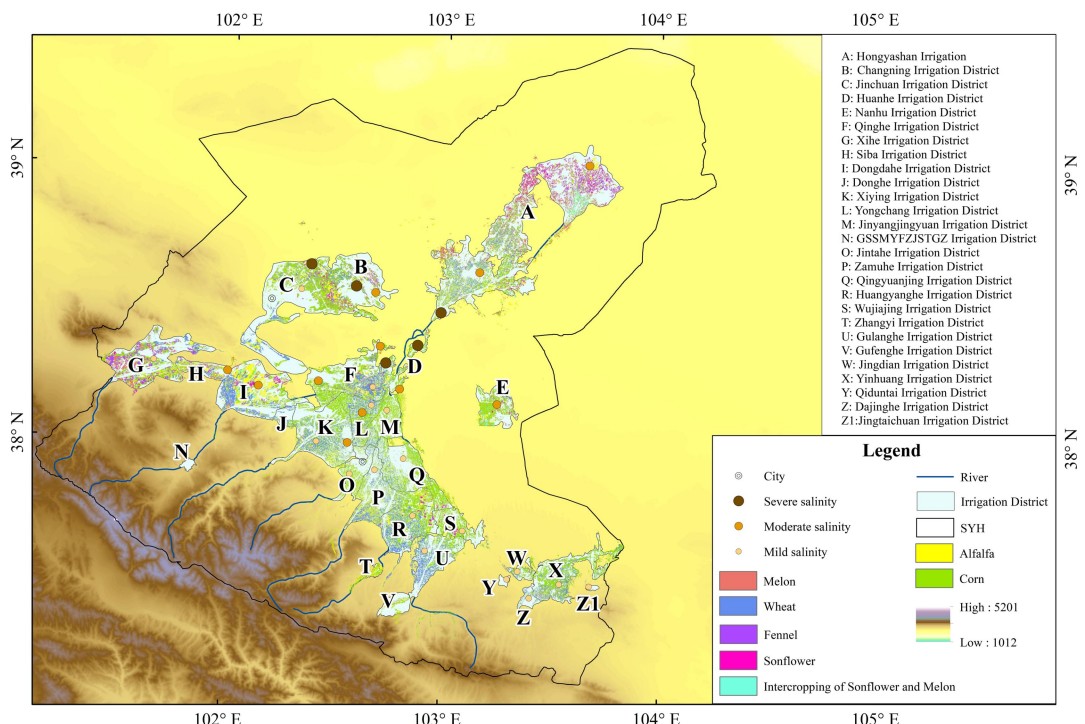

**Figure 10.** Distribution of irrigation areas in the Shiyang River Basin

## 4.3 Uncertainty in research.

This study analyzed soil salinization in the Shiyang River Basin using Landsat satellite data. However, due to the inherent uncertainties of satellite data, the results may have certain limitations. Although satellites can provide multispectral data, the spectral resolution is relatively low, and atmospheric correction issues may also affect data accuracy, posing challenges for identifying soil salinization (Vicente-Serrano et al., 2008; Vanonckelen et al., 2013). Landsat has a revisit cycle of 16 days, which can be further extended by climatic effects during certain seasons, significantly limiting seasonal monitoring of the region. Additionally, the selection and quantity of training data directly affect the accuracy of supervised classification. An accuracy assessment of the supervised classification results revealed classification accuracies of 89.40%, 88.37%, 89.80%, 99.52%, and 96.83% for the years 2002, 2007, 2012, 2017, and 2022, respectively, with kappa coefficients of 0.82, 0.81, 0.82, 0.99, and 0.95. However, due to the limitations of sampling size and satellite data, the identification of mildly saline-alkaline land is slightly less effective compared to other types of land, which requires further improvement in future work. Because soil salinization is influenced by multiple interacting factors such as climate and irrigation,

single-satellite data alone struggle to fully capture the variation of all environmental

components. Future research will expand data sources by integrating field

measurements, meteorological records, and irrigation information to obtain more

comprehensive or higher-resolution multi-source fusion data. Our systematic soil

salinity monitoring for this basin began in 2019, which represents a limited timeframe

that prevents us from comprehensively validating remote sensing interpretation results

using long-term soil physicochemical parameter data. Consequently, current accuracy

assessments primarily relies on field-verified sample points collected between 2019

and 2024, which somewhat constrains our ability to verify the long-term dynamic

processes of saline land changes. Nevertheless, these validation points still provide

important ground truth references for remote sensing monitoring results. Moreover,

the application of deep learning models for image classification and feature extraction

could deepen our understanding of the driving mechanisms behind soil salinization

distribution, thereby improving the applicability of such findings in hydrology and

soil management.

**5 Conclusion**

This study quantified soil salinization changes and its impacts on water cycle

mechanisms in the Shiyang River Basin of Northwest China's arid region from

2002-2022 using remote sensing data. The basin's salinization area showed overall

small variations, but salinity gradually intensified from southwest to northeast. The

salinity degree increased in the following ways: moderate salinization expanded in the

central corridor plain area, while severe salinization increased in the northern hills and

oasis-desert transition zone. Salinization was particularly severe around reservoirs,

certain agricultural lands, and ecological water input areas, primarily caused by

groundwater evaporation near reservoirs, agricultural irrigation evaporation, and

downstream ecological water input evaporation. However, reservoir anti-seepage

treatment measures can significantly reduce salinization problems arising from

groundwater level rises around reservoirs. The leaching effect of irrigation water

lowered salt levels in oasis irrigation districts, but salt concentrations continued to rise

in the agricultural periphery. Inappropriate land management seriously affected basin

soils, with extremely high risks of agricultural lands, grasslands, and wastelands converting to saline lands. This research will provide more scientific basis for agricultural and water resource management in the basin.

**Conflict of Interest Statement**

The authors declare no conflicts of interest.

**Author contributions statement**

Guofeng Zhu and Gaojia Meng conceived the idea of the study; Yinying Jiao, Dongdong Qiu and Yuhao Wang analyzed the data; Rui Li, Longhu Chen and Qinqin Wang participated in the drawing; Gaojia Meng wrote the paper; Siyu Lu, Enwei Huang, Jiawei Liu and Wentong Li checked and edited language. All authors discussed the results and revised the manuscript.

**Acknowledgements**

This research was financially supported by the National Natural Science Foundation of China(42371040, 41971036), Key Natural Science Foundation of Gansu Province(23JRRA698), Key Research and Development Program of Gansu Province(22YF7NA122), Cultivation Program of Major key projects of Northwest Normal University(NWNU-LKZD-202302), Oasis Scientific Research achievements Breakthrough Action Plan Project of Northwest normal University(NWNU-LZKX-202303).

**Data Availability Statement**

The 30m land use classification data for the Shiyang River Basin used in this study are available in the public domain (https://doi.org/10.5281/zenodo.4417810); Landsat series data were obtained from Earth Explorer service (https://earthxplorer. usgs.gov).

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
