# Peer review of "Soil salinity patterns reveal changes in the water cycle of inland river basins in arid zones"

_Hydrology and Earth System Sciences, 2024_

## Referee Comment (RC1)

**Journal: Hydrology and Earth System Sciences**

Title: Soil salinity patterns reveal changes in the water cycle of inland river basins in arid zones

https://doi.org/10.5194/hess-2024-76

**General comments**

This paper analyze the changes in soil salinity in the Shiyang River Basin using remote sensing and observation data from 2002 to 2022 and attempt to explore the impacts of the soil salinity changes on water conservancy projects, farmland, irrigation, and climate change. The scope of the study fits well with the journal's theme of the study of the spatial and temporal characteristics of the global water resources.

However, although the title claims that Soil salinity patterns reveal changes in the water cycle, the results section and the subsequent analysis do not justify this claim. The results section only shows the results relating to the evolution (trend of increase or decrease) of the surfaces in the different severity ranges considered, without detailing the specific methodology used to obtain them or the assessment of the accuracy of these results. In the discussion section, it is not put in context with the existing literature and appears to be a continuation of the results. Moreover, the conclusions on the impact on the water cycle are not justified by the results presented. After reviewing the manuscript and based on these comments, I recommend that the manuscript be reconsidered after a **major revision** to address the identified shortcomings.

**Specific comments**

The introduction section should expand the study of the use of remote sensing in salinity analysis, and coordinate with the new discussion section that will need to be written to compare with previous works in this field. The novelty of the article should be emphasized, especially in the Introduction.

The methodology is described very briefly 162 to 174, within the Data section, but needs to be reorganized and considerably expanded:

- All data sources should be grouped together in a single section, in order to have a better overview of the whole set of data used: satellite images, land use maps, land use maps, etc.
- Line 159 refers to a DEM but does not indicate its source.
- Line 152, please clarify if you are referring to satellite images.
- Line 153, summer images are the least likely to have clouds, but could you indicate why summer images are the best for measuring salinity?.
- Line 155, pre-processing of the image (radiometric correction) is not "improving the image quality", it is a necessary step in the process.
- Line 160, The workflow in figure 2 helps in understanding the steps but should also be explained in the body of the text.
- Line 164, it is mentioned that "In the spectral region, saline soils have higher reflectance than non-saline soils" given the importance of this aspect, it should be expanded and justified with other previous works.
- Line 165 Indicate how many control points have been used for each salinity range studied, detail their spatial location, check if there is a balance in relation to the

irrigation areas studied, etc. It should also be indicated how they have been divided into train-test-validation for accuracy control.

- Line166 indicates that the band combinations of remote sensing imagery most suitable for saline soil extraction have been adjusted. However, no further information is given on how this has been done. This part of the methodology is of vital importance in the results obtained and should be completed extensively. What alternative band combinations have been used? Which are the ones that work best and on what criteria are they based to measure it?
- Line 168 should indicate the supervised classification method that has been used, taking into account the importance of these results in the conclusions that are intended to be drawn in this study, the way in which the accuracy of the classifications has been evaluated should be described in an exhaustive manner.
- The sentence on lines 169 to 174 "In this process, the slope data, image texture features….". It is a very long sentence, and the meaning is not clear, it should be rewritten in a clearer way.
- Line 172, the concept of field sample points appears for the first time, but it is not clear whether they refer to those taken on satellite images or are field work. The location and number of these sample points should also be clarified.
- Line 180 The criteria used to define low, moderate and severe salinity ranges should be explained in detail in the method section. Given that the results of the article show almost exclusively the area classified in each severity range, it is considered of vital importance to explain how these areas have been obtained and how the accuracy of these results has been assessed.

Regarding the results and discussion sections, it seems that the division of the two sections is not clear, especially in the discussion section, as this section shows new results and new figures and does not compare the results obtained with other previous studies. In fact, the discussion section is a continuation of the results, in the entire discussion section only two papers are cited, (Thorslund et al., 2021 and Jägermeyr et al., 2017), which is a clear sign that it is not a discussion per se, but a continuation of the presentation of the results, and what is more, conclusions are observed that are not justified by the results presented.

- The analysis in the Temporal changes in soil salinization section should be rewritten to describe what is observed in the graphs, and it is useful to indicate the figure a, b, c, etc., to which you refer in each case.
- Lines 216 – 217: "*From 2002 to 2022, the overall salinized area of the basin shows an increasing trend, with an average annual growth rate of 1881.9hm2/a*". However, in fig 4a, the grey bar does not so clearly present this growth, in fact, from 2017 to 2022 there is a decrease in overall terms.
- Lines 214 and 218. The results are referred to as heavy salinization, the same nomenclature "severe" should be used.
- Lines 217 to 219:"*shows an increasing trend*", however it is maintained over time until the increase in the last period.
- Trends are analyzed by administrative counties, but the characteristics of each of them are not described: e.g. soil types, cultivated area, natural areas, semi-desertic areas, etc. Why is it analyzed by administrative terms and not for example by cluster of territory with similar characteristics? What are the factors that determine the response of the soil to the increase or decrease in salinity?

- The description of the results in the irrigation districts, line 331 onwards, should be done graphically, and not with the names of the 27 districts because it is very confusing.
- line 257, What do they mean when they talk about *transfer of water for irrigation and the rise in the water table caused by the foreign water*? Figure 5 seems key in the study but appears for the first time at the end.
- There is no relationship between conclusion (3) and the results shown and actually obtained in this document: *the regional salinization problem is more prominent as a result of the rise in the groundwater level around the reservoirs, the evaporation from the irrigation of the agricultural fields, and the evaporation from the downstream ecological water conveyance*. It should be completed and clarified. What data and results analyzed in this work support this conclusion?
- Line 370, in the statement: *The relationship between salinity and irrigation is apparent: over-irrigation increases the concentration of salts in the soil, leading to salinity problems*. It should be clarified and related to which of the irrigation areas studied have over-irrigation and relate this aspect to the specific results obtained for these areas, in order to verify if what is stated is really what is happening in these areas.

As mentioned above, there are conclusions that are not justified by the results presented:

- Lines 381- 383, The following is stated: *Farmland, grassland, and wasteland are at the most significant risk of being converted into saline soils, challenging farmland management*. However, at no point in the article are the different types of crops analyzed by irrigation areas nor are results grouped into these categories of crop types provided.
- The conclusions presented in lines 384 to 394 are not supported by the information and results presented in this work.

**Technical corrections**

In figure 1:

- I would recommend adding a more general situation map of the country, adapting the legends of the rivers, and clarifying why the lines of the channels are cut; it seems that there is no connection to the hydrological network.
- Fig 1b is within the desert zone, however they look like crop areas.
- Is Fig 1c in 3D? It is not clear what is being represented.

In figure 3:

- In the figure, unnamed counties appear; if they are part of the studied basin area, they should be named. Clarify if they are ignored because the results do not show salinity in these areas.
- It is difficult to follow the presentation of the results showed in this figure in the body of the text.
- The temporal evolution of salinity did not look very good to the naked eye in figures 3a to 3e. In addition to the figures presented, a single figure that represents, for each pixel, the relative temporal evolution throughout the time series (increase or decrease) would help in presenting the results.

In figure 4, taking into account that results are represented with a spatial component, it is recommended to represent these results on a map with their spatial distribution.

The Figure 5 is not cited in the text.

In Figure 6, the results shown are not located correctly in space, it would be helpful to superimpose the scale, north, etc. on the aerial image.

Figure 7, in this figure it seems that there is continuity of the channels and irrigation areas. Is that why they appear cut off in Fig. 1? This figure can be greatly improved, for example with maps of crop types in each area, soil types, etc., with data that provides information on the context of the space studied.

Other minor comments:

- It is detected that a space is missing after the period and it often happens frequently throughout the document (e.g. 136, 142, 150, etc.)
- line 144, It seems that the phrase "Detect more subtleties" is repetitive of the previous one.
- line 217, in units $hm^2/a$ What is the meaning of a? annual?

---

## Referee Comment (RC2)

**Journal:** Hydrology and Earth System Sciences

**Title:** Soil salinity patterns reveal changes in the water cycle of inland river basins in arid zones

**Authors:** Gaojia Meng, Guofeng Zhu, Yinying Jiao, Dongdong Qiu, Yuhao Wang, Siyu Lu, Rui Li, Jiawei Liu, Longhu Chen, Qinqin Wang, Enwei Huang, and Wentong Li

https://doi.org/10.5194/hess-2024-76

**General comments:**
This study examined the soil salinity patterns reveal changes in the water cycle of inland river basins in arid zones. The results contribute towards a better conceptualisation of soil salinity patterns in inland river basins in arid zones China. The topic of these article is appropriate for the journal of Hydrology and Earth System Sciences. However, the manuscript still has many shortcomings and needs to be further improved. Therefore, I would recommend the manuscript for major revisions.

**Specific comments:**
Abstract:
Line 6: Add to the abstract the country where the Shiyang River Basin is located. It will help to make it easier to find the article during following reviews and meta-analyses.

Introduction:
The Introduction is conceived too generally and in a broad context. I recommend focusing more on changing soil salinity in the context of landuse management.

Materials and Methods:
Lines 104 – 108: General description of climatic conditions at the sites, include specific ranges of long-term meteorological variables (air temperatures, precipitation) for the period of the last 30 years (1991 – 2020).

Lines 108 – 111: Add bedrock for all four geomorphological units.

Lines 111 – 116: Change the soil classification to one of the international classification systems, e.g. *"World reference base for soil resources 4th edition (2022)"*. Please add also estimated soil texture and water permeability of these soils based on available references.

Lines 104 – 116: Add main landcover and land type for all geomorphological units.

Lines 104 – 116: If information is available, add the groundwater level for all geomorphological units.

Lines 162 – 174: The methodology of data processing and synthesis is too concisely conceived and requires expansion and logical continuity.

Line 172: For field sampling points, add the number of points, sampling depth and design of soil sampling at sites, e.g. regular grid, irregular grid or random sampling. If necessary, add a map with field sampling points.

Results:
Line 180: Nowhere in the methodology is there an explanation of the classification of mild, moderate, and severe soil salinization. Are there differences in the various salt concentrations in the soil (µS/cm) for these three categories? Please, add this classification to Material and Methods chapter.

Lines 211 – 212: I would recommend that the results in the previous subsection *"3.1 Spatial distribution of soil salinisation"* should also be described according to administrative boundaries so that the results are clear. The division of areas according to administrative boundaries should also be descripted in the methodology.

Line 217: Why did you use the unit $hm^2/a$? It would be more accepted to use $km^2/a$, which will be understood by more people from different country. For example, in Europe, the $hm^2$ unit is used rarely. What does "/a" mean? Is it a change in a year?

Discussion:
In my opinion, the discussion chapter seems to me to be a continuation of the results chapter, with detail on river water transfer, Red Bluff Mountain Reservoir, and irrigation and salinization. I believe that research uncertainty and research perspectives need to be added to the discussion section so that manuscript studies can be made more complete. The chapter lacks a clear critical view of the issue and a confrontation of the results with other science results. In this chapter, I expect critical evaluation of the author's results, a follow-up to the current state of knowledge in the issue of soil salinization in a global context. At the same time, it is appropriate for critics to evaluate the limits, weakness of this study (methodical, interpretative, etc.) and, on the contrary, also the study strengths that contribute to the understanding of knowledge in the issue of global soil salinations.

Lines 305 – 308: What is the source of this information? Were they observed directly by this study, or were these results from some previous study? If yes, please add a reference.

Conclusion:
In this chapter, there are unfounded conclusions that were not presented in the results or discussed subsequently. Lines 381 – 383: "*Farmland, grassland, and wasteland are at the most significant risk of being converted into saline soils, challenging farmland management.*" However, landuse was not analysed in this study, and the extent of changes in landuse during the studied period is not study.

**Technical corrections:**
In Figure 1: Please add compass rose to legend. The Figure 5 is not cited in the text.

Lines 155 – 160: Please, add references (sources) to the used software (ENVI and GIS software) and to the used digital data (Slope and DEM data).

There are occasional typos in the entire text, especially in some places there are missing spaces after the end of the sentence. Please check all text.

I hope that my comments will not discourage you and will contribute to the improvement of this work. Thank you for allowing me to review this manuscript of your article.

---

## Author Comment (AC1)

Response to Reviewer#2:

Your valuable insights have significantly contributed to enhancing the quality of our manuscript. I feel extremely honored to receive such positive and constructive feedback from you. I genuinely appreciate every thoughtful suggestion you've provided and will address each one of them with utmost care, offering detailed responses in return.

In the revised manuscript, we have meticulously restructured and refined the logic and content of the abstract, introduction, discussion, and image sections. **The revisions in the manuscript are indicated using red font.** Below is a comprehensive overview of the modifications we have made:

**General comments:**

This study examined the soil salinity patterns reveal changes in the water cycle of inland river basins in arid zones. The results contribute towards a better conceptualisation of soil salinity patterns in inland river basins in arid zones China. The topic of these article is appropriate for the journal of Hydrology and Earth System Sciences. However, the manuscript still has many shortcomings and needs to be further improved. Therefore, I would recommend the manuscript for major revisions.

**Respond:** Thank you very much for your insights. To provide more comprehensive content, we have diligently reviewed and refined the abstract, introduction, literature review, research methods,results, discussion and conclusion, aiming to enhance the depth of our research focus.

**Specific comments:**

Abstract:

Line 6: Add to the abstract the country where the Shiyang River Basin is located. It will help to make it easier to find the article during following reviews and meta-analyses.

**Respond:** Thank you for your suggestion. We have added it to the original text. As follows:

Based on remote sensing and observation data, this study quantitatively analyzed

the changes in soil salinization in the Shiyang River Basin, an arid region in Northwest China, from 2002 to 2022. It also explored the impact of hydraulic engineering and farmland irrigation on soil salinization.

Introduction:

The Introduction is conceived too generally and in a broad context. I recommend focusing more on changing soil salinity in the context of landuse management.

**Respond:**We have reorganized the structure of the introduction section and added relevant research, as shown below:

[revised manuscript text omitted]

Zhang and Huang, 2019; Lotfollahi et al., 2023). Saline soils show absorption peaks in the visible band, and there is a positive correlation between their soil reflectance and soil salinity. In world-scale soil salinity studies, researchers have used machine learning methods to monitor the dynamics of soil surface salinity over the past four decades (Hassani et al., 2020) and ML algorithms to predict soil salinity in the 21st century in the context of global climate change (Has-sani et al., 2021). It was found that the salt-affected areas were mainly distributed in arid and semi-arid regions, significantly more severe in northwestern China (Li et al., 2014). The risk challenge of soil salinization is further increased in arid and semi-arid regions of China due to their special climatic conditions, which are influenced by irrigation, drainage, and ecological water transport (Wang et al., 2012; Miguel et al., 2013). The temporal and spatial relationship between soil salinization and groundwater decline exacerbates the regional water-salt imbalance. Irrigated agriculture carries salts into the groundwater layers, leading to increased groundwater salinity and resulting in soil salinization in irrigation areas (Foster et al., 2018). Furthermore, as more land is converted to farmland, increased irrigation leads to salt accumulation. The overexploitation of land resources has had a significant and lasting impact on soil salinization (Wang et al., 2013; Yin et al., 2021).

The Shiyang River Basin, located in the arid region of Northwest China, is a typical inland river basin where soil salinization is a prominent issue closely linked to factors such as hydraulic engineering and irrigation activities. Therefore, assessing the distribution of soil salinization in this basin is crucial for understanding how natural and human activities impact soil salinization in arid areas. In this study, we aim to address the following questions: (1) Quantitatively analyze the degree of salinization in the Shiyang River Basin and reveal its spatial and temporal distribution characteristics; (2) analyze the impacts of water cycle changes on salinization. The study's results will help clarify the impact of the water cycle on soil salinization in the inland river basin and provide a scientific basis for agricultural development, ecological construction, and water resource use planning in the arid zone.

Materials and Methods:

Lines 104 – 108: General description of climatic conditions at the sites, include specific ranges of long-term meteorological variables (air temperatures, precipitation) for the period of the last 30 years (1991 – 2020).

**Respond:** Thank you for your suggestion. We have described the meteorological variables of the study area according to the geomorphological units, as shown below:

The bedrock of the southern Qilian Mountains consists of metamorphosed sandstones and volcanic rocks, with soil types including Cryosols, Leptosols, and Phaeozems. The land is primarily forest and grassland, with annual precipitation of 300-600mm, evaporation rates of 700-1200mm, and the groundwater level is 50-200 meters below the surface. The central corridor plain features bedrock composed of schist and slate, with soil types including Gypsisols, Calcisols, and Solonchaks. The land use is primarily agricultural, with annual precipitation of 150-300mm, evaporation rates of 1300-2000mm, and the groundwater level is 50 meters below the surface. The northern low hills and deserts have predominantly igneous bedrock, and soils consisting of Arenosols, Leptosols, and Solonchaks. The landscape is barren, with annual precipitation below 150mm, evaporation rates of 2000-3000mm, and the groundwater level is 30 meters below the surface.

Lines 108 – 111: Add bedrock for all four geomorphological units.

**Respond:** We have added bedrock information for the four geomorphological units, as shown below:

The bedrock of the southern Qilian Mountains consists of metamorphosed sandstones and volcanic rocks, with soil types including Cryosols, Leptosols, and Phaeozems. The land is primarily forest and grassland, with annual precipitation of 300-600mm, evaporation rates of 700-1200mm, and the groundwater level is 50-200 meters below the surface. The central corridor plain features bedrock composed of schist and slate, with soil types including Gypsisols, Calcisols, and Solonchaks. The land use is primarily agricultural, with annual precipitation of 150-300mm, evaporation rates of 1300-2000mm, and the groundwater level is 50 meters below the

surface. The northern low hills and deserts have predominantly igneous bedrock, and soils consisting of Arenosols, Leptosols, and Solonchaks. The landscape is barren, with annual precipitation below 150mm, evaporation rates of 2000-3000mm, and the groundwater level is 30 meters below the surface.

Lines 111 – 116: Change the soil classification to one of the international classification systems, e.g. "World reference base for soil resources 4th edition (2022)". Please add also estimated soil texture and water permeability of these soils based on available references.

**Respond:** Thank you for your suggestion. We have consulted the "World reference base for soil resources 4th edition (2022)", and revised the soil classification in the original text as follows:

The bedrock of the southern Qilian Mountains consists of metamorphosed sandstones and volcanic rocks, with soil types including Cryosols, Leptosols, and Phaeozems. The land is primarily forest and grassland, with annual precipitation of 300-600mm, evaporation rates of 700-1200mm, and the groundwater level is 50-200 meters below the surface. The central corridor plain features bedrock composed of schist and slate, with soil types including Gypsisols, Calcisols, and Solonchaks. The land use is primarily agricultural, with annual precipitation of 150-300mm, evaporation rates of 1300-2000mm, and the groundwater level is 50 meters below the surface. The northern low hills and deserts have predominantly igneous bedrock, and soils consisting of Arenosols, Leptosols, and Solonchaks. The landscape is barren, with annual precipitation below 150mm, evaporation rates of 2000-3000mm, and the groundwater level is 30 meters below the surface.

Lines 104 – 116: Add main landcover and land type for all geomorphological units.

**Respond:** We have added the main land cover types for the four geomorphological units, as shown below:

The bedrock of the southern Qilian Mountains consists of metamorphosed sandstones and volcanic rocks, with soil types including Cryosols, Leptosols, and Phaeozems. The land is primarily forest and grassland, with annual precipitation of

300-600mm, evaporation rates of 700-1200mm, and the groundwater level is 50-200 meters below the surface. The central corridor plain features bedrock composed of schist and slate, with soil types including Gypsisols, Calcisols, and Solonchaks. The land use is primarily agricultural, with annual precipitation of 150-300mm, evaporation rates of 1300-2000mm, and the groundwater level is 50 meters below the surface. The northern low hills and deserts have predominantly igneous bedrock, and soils consisting of Arenosols, Leptosols, and Solonchaks. The landscape is barren, with annual precipitation below 150mm, evaporation rates of 2000-3000mm, and the groundwater level is 30 meters below the surface.

Lines 104 － 116: If information is available, add the groundwater level for all geomorphological units.

**Respond:** After consulting the "Annual Report of Groundwater Level Monitoring for China's Geological Environment" and relevant research papers, we obtained the groundwater level ranges for the geomorphological units in the study area. This information has been added to the original text, as shown below:

The bedrock of the southern Qilian Mountains consists of metamorphosed sandstones and volcanic rocks, with soil types including Cryosols, Leptosols, and Phaeozems. The land is primarily forest and grassland, with annual precipitation of 300-600mm, evaporation rates of 700-1200mm, and the groundwater level is 50-200 meters below the surface. The central corridor plain features bedrock composed of schist and slate, with soil types including Gypsisols, Calcisols, and Solonchaks. The land use is primarily agricultural, with annual precipitation of 150-300mm, evaporation rates of 1300-2000mm, and the groundwater level is 50 meters below the surface. The northern low hills and deserts have predominantly igneous bedrock, and soils consisting of Arenosols, Leptosols, and Solonchaks. The landscape is barren, with annual precipitation below 150mm, evaporation rates of 2000-3000mm, and the groundwater level is 30 meters below the surface.

Lines 162 – 174: The methodology of data processing and synthesis is too concisely conceived and requires expansion and logical continuity.

**Respond:** Thank you for your suggestion. We have rewritten this section as follows:

This article selects the years 2002, 2007, 2012, 2017, and 2022 as the study periods, with four satellite remote sensing images chosen for each period to cover the entire study area. Preference is given to downloading high-quality satellite remote sensing images from the summer of each year, with cloud cover less than 1%, as this is more conducive to identifying the salinity and alkalinity levels of the soil (Allbed & Kumar, 2013). For the subsequent remote sensing inversion of salinity and alkalinity, preliminary preprocessing of the images in ENVI5.3 software is necessary(Source:https://www.l3harrisgeospatial.com/Software-Technology/ENVI), including steps such as radiometric calibration, atmospheric correction, image fusion, image mosaicking, and image clipping. Based on the natural attributes of the soil in the study area, auxiliary data, and field survey conditions, we use high-resolution images from Google Maps to select interpretation markers for mild, moderate, and severe saline-alkaline land and other land types. Next, we adjust the band combination of satellite remote sensing images to be most suitable for extracting saline-alkaline land (Khan et al., 2005; Jia et al., 2024). Using the Normalized Difference Salinity Index (NDSI), slope data, and texture features as references, we employ a Support Vector Machine (SVM) algorithm for supervised classification to identify the distribution of saline-alkaline land in the study area. The formula is as follows:

$$\min_{w,\, b,\, \xi_i} \left( \frac{1}{2} \|w\|^2 + C \sum_{i=1}^{n} \xi_i \right) \tag{1}$$

$$y_i \left( w \cdot x_i \right) \geq 1 - \xi_i, \xi_i \geq 0, i = 1, \ldots, n \tag{2}$$

In the formula, w represents the weight vector, which defines the direction of the hyperplane; b is the bias term, defining the offset of the hyperplane; $\xi_i$ is the slack variable, which increases the robustness of the model; C is the regularization parameter, balancing the model complexity and training error; $y_i$ is the label of data point i, commonly used to define a hyperplane.

Finally, the accuracy of the supervised classification results is evaluated using the confusion matrix method, including overall classification accuracy, Kappa coefficient, etc. The data processing flow is shown in Figure 2.

Line 172: For field sampling points, add the number of points, sampling depth and design of soil sampling at sites, e.g. regular grid, irregular grid or random sampling. If necessary, add a map with field sampling points.

**Respond:**Thank you for pointing out the lack of clarity in our description. The sampling points we refer to are based on the natural attributes of the soil in the study area, auxiliary data, and field research conditions. These points are established as interpretation markers in areas classified as slightly, moderately, and severely saline-alkali land, as well as in other land types. We have rewritten this part as follows:

This article selects the years 2002, 2007, 2012, 2017, and 2022 as the study periods, with four satellite remote sensing images chosen for each period to cover the entire study area. Preference is given to downloading high-quality satellite remote sensing images from the summer of each year, with cloud cover less than 1%, as this is more conducive to identifying the salinity and alkalinity levels of the soil (Allbed & Kumar, 2013). For the subsequent remote sensing inversion of salinity and alkalinity, preliminary preprocessing of the images in ENVI5.3 software is necessary(Source:https://www.l3harrisgeospatial.com/Software-Technology/ENVI), including steps such as radiometric calibration, atmospheric correction, image fusion, image mosaicking, and image clipping. Based on the natural attributes of the soil in the study area, auxiliary data, and field survey conditions, we use high-resolution images from Google Maps to select interpretation markers for mild, moderate, and severe saline-alkaline land and other land types. Next, we adjust the band combination of satellite remote sensing images to be most suitable for extracting saline-alkaline land (Khan et al., 2005; Jia et al., 2024). Using the Normalized Difference Salinity Index (NDSI), slope data, and texture features as references, we employ a Support Vector Machine (SVM) algorithm for supervised classification to identify the distribution of saline-alkaline land in the study area. The formula is as follows:

$$\min_{w, \, b, \, \xi_i} \left( \frac{1}{2} \|w\|^2 + C \sum_{i=1}^{n} \xi_i \right) \tag{1}$$

$$y_i(w \cdot x_i) \geq 1 - \xi_i, \xi_i \geq 0, i = 1, ..., n \qquad (2)$$

In the formula, w represents the weight vector, which defines the direction of the hyperplane; b is the bias term, defining the offset of the hyperplane; $\xi_i$ is the slack variable, which increases the robustness of the model; C is the regularization parameter, balancing the model complexity and training error; $y_i$ is the label of data point i, commonly used to define a hyperplane.

Finally, the accuracy of the supervised classification results is evaluated using the confusion matrix method, including overall classification accuracy, Kappa coefficient, etc. The data processing flow is shown in Figure 2.

Results:

Line 180: Nowhere in the methodology is there an explanation of the classification of mild, moderate, and severe soil salinization. Are there di\erences in the various salt concentrations in the soil ( μ S/cm) for these three categories? Please, add this classification to Material and Methods chapter.

**Respond:**Thank you for your suggestion. We have added the following to the original text:

This article selects the years 2002, 2007, 2012, 2017, and 2022 as the study periods, with four satellite remote sensing images chosen for each period to cover the entire study area. Preference is given to downloading high-quality satellite remote sensing images from the summer of each year, with cloud cover less than 1%, as this is more conducive to identifying the salinity and alkalinity levels of the soil (Allbed & Kumar, 2013). For the subsequent remote sensing inversion of salinity and alkalinity, preliminary preprocessing of the images in ENVI5.3 software is necessary(Source:https://www.l3harrisgeospatial.com/Software-Technology/ENVI), including steps such as radiometric calibration, atmospheric correction, image fusion, image mosaicking, and image clipping. Based on the natural attributes of the soil in the study area, auxiliary data, and field survey conditions, we use high-resolution images from Google Maps to select interpretation markers for mild, moderate, and severe saline-alkaline land and other land types. Next, we adjust the band combination of

satellite remote sensing images to be most suitable for extracting saline-alkaline land (Khan et al., 2005; Jia et al., 2024). Using the Normalized Difference Salinity Index (NDSI), slope data, and texture features as references, we employ a Support Vector Machine (SVM) algorithm for supervised classification to identify the distribution of saline-alkaline land in the study area. The formula is as follows:

$$\min_{w, b, \xi_i} \left( \frac{1}{2} \|w\|^2 + C \sum_{i=1}^{n} \xi_i \right) \tag{1}$$

$$y_i (w \cdot x_i) \geq 1 - \xi_i, \xi_i \geq 0, i = 1, ..., n \tag{2}$$

In the formula, w represents the weight vector, which defines the direction of the hyperplane; b is the bias term, defining the offset of the hyperplane; $\xi_i$ is the slack variable, which increases the robustness of the model; C is the regularization parameter, balancing the model complexity and training error; $y_i$ is the label of data point i, commonly used to define a hyperplane.

Finally, the accuracy of the supervised classification results is evaluated using the confusion matrix method, including overall classification accuracy, Kappa coefficient, etc. The data processing flow is shown in Figure 2.

Lines 211 – 212: I would recommend that the results in the previous subsection "3.1 Spatial distribution of soil salinisation" should also be described according to administrative boundaries so that the results are clear. The division of areas according to administrative boundaries should also be described in the methodology.

**Respond:**Thank you for your suggestion. Considering that describing based on natural landform units can make the results clearer, we have rewritten this section as follows:

Remote sensing inversion of salinization in the Shiyang River Basin from 2002 to 2022 was carried out based on the selected samples of mild, moderate, and severe soil salinization (Fig. 3). The results showed that the salinization of the basin gradually increased from upstream to downstream, especially in the downstream of the basin near Qingtu Lake, where the salinization of the soil was the most serious.

From the perspective of natural landform division, the salt-accumulating areas of the Shiyang River Basin are widely distributed across the central corridor plains, northern low mountains, hills, and desert areas. In the central corridor plains, soil salinization is mainly characterized by mild and moderate salinization. Moderate saline soils are primarily concentrated in the oasis farmland irrigation areas on both sides of the river, with a few plots transforming into severe saline soils. The area of moderate saline land expanded significantly in 2012, with growth areas located in the central part of the plains. Mild saline lands are scattered and cover a smaller area. In 2012, a large number of new mildly saline plots emerged in the western part of the central corridor plains, and the area of mild saline land increased in the southeast. By 2022, mild salinization in these areas had improved to some extent. In the northern low mountains, hills, and desert areas, soil salinization is mainly characterized by moderate and severe salinization, with the area and extent far exceeding those of the central corridor plains. Moderate saline lands are mostly located in semi-desert areas and outside irrigation zones, especially at the end of the Shiyang River Basin, where downstream salt accumulation is prevalent, resulting in a concentration of heavily saline lands. In contrast, mildly saline lands are less common and scattered.

Line 217: Why did you use the unit hm2 /a? It would be more accepted to use $km^2$ /a, which will be understood by more people from di\erent country. For example, in Europe, the hm2 unit is used rarely. What does "/a" mean? Is it a change in a year?

**Respond:**As you mentioned, the use of $hm^2$ /a is quite rare, so we have replaced all instances of $hm^2$ with $km^2$ throughout the text. The "/a" indicates the average annual change, representing a trend of change.

Discussion:

In my opinion, the discussion chapter seems to me to be a continuation of the results chapter, with detail on river water transfer, Red Blu\ Mountain Reservoir, and irrigation and salinization. I believe that research uncertainty and research perspectives need to be added to the discussion section so that manuscript studies can be made more complete. The chapter lacks a clear critical view of the issue and a confrontation of the results with other science results. In this chapter, I expect critical

evaluation of the author's results, a follow-up to the current state of knowledge in the issue of soil salinization in a global context. At the same time, it is appropriate for critics to evaluate the limits, weakness of this study (methodical, interpretative, etc.) and, on the contrary, also the study strengths that contribute to the understanding of knowledge in the issue of global soil salinations.

**Respond:**Thank you for your suggestion. We have improved the discussion section by integrating it with existing studies and added an uncertainty analysis in section 4.3.

Lines 305 – 308: What is the source of this information? Were they observed directly by this study, or were these results from some previous study? If yes, please add a reference.

**Respond:**The groundwater EC (Electrical Conductivity) values were obtained through our measured data, we did not list the data separately.

Conclusion:

In this chapter, there are unfounded conclusions that were not presented in the results or discussed subsequently. Lines 381 – 383: "Farmland, grassland, and wasteland are at the most significant risk of being converted into saline soils, challenging farmland management." However, landuse was not analysed in this study, and the extent of changes in landuse during the studied period is not study.

**Respond:**Regarding the question you mentioned, we processed the land use type data of the Shiyang River Basin in our preliminary work and calculated the land use transfer matrix for the years 2002 to 2022. In the statistics on land type changes, alkali-saline land converted from farmland accounted for 3.07% of the alkali-saline land area in 2022, conversion from grassland accounted for 6.23%, and conversion from wasteland had the largest share at 11.27%. Therefore, we put forward this view in the conclusion section. We have now added the relevant calculation results to section 4.2 of the original paper.

Technical corrections:

In Figure 1: Please add compass rose to legend. The Figure 5 is not cited in the text.

**Respond:**We have added a north arrow to Figure 1 and have included a reference to

Figure 5 in section 4.1.

[Figure]

Ecological water transfer mainly refers to the Jingdian water transfer (i.e., water diversion from the Yellow River), which regulates the irrigation water use pattern by transferring water to Qingtuhu Lake through the Hongya Mountain Reservoir (Fig. 5).

Lines 155 – 160: Please, add references (sources) to the used software (ENVI and GIS software) and to the used digital data (Slope and DEM data).

**Respond:**Thank you for your suggestion. We have added the sources for the software and numerical data in the original text as follows:

Landsat data is available from the Earth Explorer service (https://earthexplorer.usgs.gov), which provides surface reflectance every 16 days with a spatial resolution of 30 meters.

This article obtained the 30m land use data product for the Shiyang River Basin from 2002 to 2022, which is available in the public domain at https://doi.org/10.5281/zenodo.4417810 (Yang and Huang, 2022).

The Digital Elevation Model (DEM) data is ASTER GDEM data jointly developed by Japan's METI and the U.S. NASA, distributed to the public for free with a resolution of 30m. This data can be downloaded at http://reverb.echo.nasa.gov/reverb/. Slope data is calculated from the DEM data.

For the subsequent remote sensing inversion of salinity and alkalinity, preliminary preprocessing of the images in ENVI5.3 software is necessary(Source:https://www.l3harrisgeospatial.com/Software-Technology/ENVI).

There are occasional typos in the entire text, especially in some places there are missing spaces after the end of the sentence. Please check all text.

**Respond:**Thank you for your suggestion. We have reviewed and revised the entire text.

---

## Author Comment (AC2)

Response to Reviewer#1:

Your valuable insights have significantly contributed to enhancing the quality of our manuscript. I feel extremely honored to receive such positive and constructive feedback from you. I genuinely appreciate every thoughtful suggestion you've provided and will address each one of them with utmost care, offering detailed responses in return.

In the revised manuscript, we have meticulously restructured and refined the logic and content of the abstract, introduction, discussion, and image sections. **The revisions in the manuscript are indicated using red font.** Below is a comprehensive overview of the modifications we have made:

**General comments:**

This paper analyze the changes in soil salinity in the Shiyang River Basin using remote sensing and observation data from 2002 to 2022 and attempt to explore the impacts of the soil salinity changes on water conservancy projects, farmland, irrigation, and climate change. The scope of the study fits well with the journal's theme of the study of the spatial and temporal characteristics of the global water resources.

However, although the title claims that Soil salinity patterns reveal changes in the water cycle, the results section and the subsequent analysis do not justify this claim. The results section only shows the results relating to the evolution (trend of increase or decrease) of the surfaces in the different severity ranges considered, without detailing the specific methodology used to obtain them or the assessment of the accuracy of these results. In the discussion section, it is not put in context with the existing literature and appears to be a continuation of the results. Moreover, the conclusions on the impact on the water cycle are not justified by the results presented. After reviewing the manuscript and based on these comments, I recommend that the manuscript be reconsidered after a major revision to address the identified shortcomings.

**Respond:**Thank you very much for your insights. To provide more comprehensive

content, we have diligently reviewed and refined the abstract, introduction, literature review, research methods,results, discussion and conclusion, aiming to enhance the depth of our research focus.

**Specific comments:**

The introduction section should expand the study of the use of remote sensing in salinity analysis, and coordinate with the new discussion section that will need to be written to compare with previous works in this field. The novelty of the article should be emphasized, especially in the Introduction.

**Respond:**We have reorganized the structure of the introduction section and added relevant research, as shown below:

[revised manuscript text omitted]

All data sources should be grouped together in a single section, in order to have a better overview of the whole set of data used: satellite images, land use maps, land use maps, etc.

**Respond:**Thank you for your suggestions. We have summarized and organized the data sources used in section2.2, including land satellite data, land use data, and digital elevation models, and indicated their sources.

• Line 159 refers to a DEM but does not indicate its source.

**Respond:**The source of the DEM is the ASTER GDEM data, jointly developed and freely distributed to the public by Japan's METI and NASA. I have added the data source to the text, as shown below:

The Digital Elevation Model (DEM) data is ASTER GDEM data jointly developed by Japan's METI and the U.S. NASA, distributed to the public for free with a resolution of 30m. This data can be downloaded at http://reverb.echo.nasa.gov/reverb/. Slope data is calculated from the DEM data.

• Line 152, please clarify if you are referring to satellite images.

**Respond:**We are referring to the Landsat series satellite imagery, and I have made clarifications in the text as shown below:

.This article selects the years 2002, 2007, 2012, 2017, and 2022 as the study periods, with four satellite remote sensing images chosen for each period to cover the entire study area.

• Line 153, summer images are the least likely to have clouds, but could you indicate why summer images are the best for measuring salinity?

**Respond:**In the identification of saline soils, the primary reason for prioritizing summer remote sensing images is that the high-temperature and dry conditions in summer promote more frequent water and salt movement. In this environment, surface evaporation is significant, salt crystallization is enhanced, and combined with the effects of irrigation, soil salinity tends to accumulate on the surface. Therefore, it is easier to identify through remote sensing images (Allbed & Kumar, 2013).

Additionally, the area around the Shiyang River is covered with snow in winter, March in spring, and November in autumn, making it difficult to effectively identify salinity.

Allbed A, Kumar L. Soil salinity mapping and monitoring in arid and semi-arid regions using remote sensing technology: a review[J]. Advances in remote sensing, 2013, 2013.

• Line 155, pre-processing of the image (radiometric correction) is not "improving the image quality", it is a necessary step in the process.

**Respond:**Thank you for pointing out the issue. We have made modifications to this section in the original text as shown below:

For the subsequent remote sensing inversion of salinity and alkalinity, preliminary preprocessing of the images in ENVI5.3 software is necessary(Source:https://www.l3harrisgeospatial.com/Software-Technology/ENVI), including steps such as radiometric calibration, atmospheric correction, image fusion, image mosaicking, and image clipping.

• Line 160, The workflow in figure 2 helps in understanding the steps but should also be explained in the body of the text.

**Respond:**We have provided a textual description of the flowchart in Figure 2 in Section 2.3, as shown below:

This article selects the years 2002, 2007, 2012, 2017, and 2022 as the study periods, with four satellite remote sensing images chosen for each period to cover the entire study area. Preference is given to downloading high-quality satellite remote sensing images from the summer of each year, with cloud cover less than 1%, as this is more conducive to identifying the salinity and alkalinity levels of the soil (Allbed & Kumar, 2013). For the subsequent remote sensing inversion of salinity and alkalinity, preliminary preprocessing of the images in ENVI5.3 software is necessary(Source:https://www.l3harrisgeospatial.com/Software-Technology/ENVI), including steps such as radiometric calibration, atmospheric correction, image fusion, image mosaicking, and image clipping. Based on the natural attributes of the soil in the study area, auxiliary data, and field survey conditions, we use high-resolution images from

Google Maps to select interpretation markers for mild, moderate, and severe saline-alkaline land and other land types. Next, we adjust the band combination of satellite remote sensing images to be most suitable for extracting saline-alkaline land (Khan et al., 2005; Jia et al., 2024). Using the Normalized Difference Salinity Index (NDSI), slope data, and texture features as references, we employ a Support Vector Machine (SVM) algorithm for supervised classification to identify the distribution of saline-alkaline land in the study area. The formula is as follows:

$$\min_{w,\, b,\, \xi_i} \left( \frac{1}{2}\|w\|^2 + C\sum_{i=1}^{n}\xi_i \right) \tag{1}$$

$$y_i\left(w \cdot x_i\right) \geq 1 - \xi_i,\, \xi_i \geq 0,\, i = 1,\ldots,n \tag{2}$$

In the formula, w represents the weight vector, which defines the direction of the hyperplane; b is the bias term, defining the offset of the hyperplane; $\xi_i$ is the slack variable, which increases the robustness of the model; C is the regularization parameter, balancing the model complexity and training error; $y_i$ is the label of data point i, commonly used to define a hyperplane.

Finally, the accuracy of the supervised classification results is evaluated using the confusion matrix method, including overall classification accuracy, Kappa coefficient, etc. The data processing flow is shown in Figure 2.

• Line 164, it is mentioned that "In the spectral region, saline soils have higher reflectance than non-saline soils" given the importance of this aspect, it should be expanded and justified with other previous works.

**Respond:** During the literature review for the article, we observed this viewpoint extensively in numerous studies: in the visible and near-infrared bands, saline soils reflect more than non-saline soils due to their lower moisture content. Therefore, we adopted this viewpoint in our research. Given its importance, we have added the relevant references in the original text, as shown below:

Metternicht G I, Zinck J A. Remote sensing of soil salinity: potentials and constraints[J]. Remote sensing of environment, 2003, 85(1): 1-20.

Farifteh J, Van der Meer F, Atzberger C, et al. Quantitative analysis of salt-affected

soil reflectance spectra: A comparison of two adaptive methods (PLSR and ANN)[J]. Remote Sensing of Environment, 2007, 110(1): 59-78.

El Harti A, Lhissou R, Chokmani K, et al. Spatiotemporal monitoring of soil salinization in irrigated Tadla Plain (Morocco) using satellite spectral indices[J]. International Journal of Applied Earth Observation and Geoinformation, 2016, 50: 64-73.

Zhang X, Huang B. Prediction of soil salinity with soil-reflected spectra: A comparison of two regression methods[J]. Scientific Reports, 2019, 9(1): 5067.

Lotfollahi L, Delavar M A, Biswas A, et al. Spectral prediction of soil salinity and alkalinity indicators using visible, near-, and mid-infrared spectroscopy[J]. Journal of Environmental Management, 2023, 345: 118854.

• Line 165 Indicate how many control points have been used for each salinity range studied, detail their spatial location, check if there is a balance in relation to the irrigation areas studied, etc. It should also be indicated how they have been divided into train-test-validation for accuracy control.

**Respond:**In this study, remote sensing images from 2002, 2007, 2012, 2017, and 2022 were selected for salinity inversion in the Shiyang River Basin. Control points were selected on Google Maps for classification, with 25 points each chosen for the four areas of interest: slightly saline-alkaline land, moderately saline-alkaline land, heavily saline-alkaline land, and others. These control points can be overlaid on the irrigation area. Finally, sample points were used to verify the accuracy of the classification results. The classification accuracies for the five years were 89.40%, 88.37%, 89.80%, 99.52%, and 96.83%, respectively, and the kappa coefficients were 0.82, 0.81, 0.82, 0.99, and 0.95.

• Line166 indicates that the band combinations of remote sensing imagery most suitable for saline soil extraction have been adjusted. However, no further information is given on how this has been done. This part of the methodology is of vital importance in the results obtained and should be completed extensively. What alternative band combinations have been used? Which are the ones that work best and on what criteria are they based to measure it?

**Respond:** Since the spectral characteristics of saline soils are quite distinct in the visible and near-infrared bands, we often use various spectral salinity indices for monitoring saline soils. Examples include the three salinity indices proposed by Khan: Brightness Index (BI), Normalized Difference Salinity Index (NDSI), and Salinity Index (SI), among which the NDSI has shown better results in research (Khan et al., 2005; Jia et al., 2024). Therefore, in the preliminary stages, we calculate the NDSI for the bands and use it as one of the features to identify and classify saline-alkaline land.

Jia P, Zhang J, Liang Y, et al. The inversion of arid-coastal cultivated soil salinity using explainable machine learning and Sentinel-2[J]. Ecological Indicators, 2024, 166: 112364.

Khan N M, Rastoskuev V V, Sato Y, et al. Assessment of hydrosaline land degradation by using a simple approach of remote sensing indicators[J]. Agricultural Water Management, 2005, 77(1-3): 96-109.

• Line 168 should indicate the supervised classification method that has been used, taking into account the importance of these results in the conclusions that are intended to be drawn in this study, the way in which the accuracy of the classifications has been evaluated should be described in an exhaustive manner.

**Respond:** Thank you for your suggestion. We have conducted an accuracy verification of the classification results, and we will include this part in the uncertainty analysis. Additionally, we have made revisions and additions to the supervised classification methods in the original text, as shown below:

we employ a Support Vector Machine (SVM) algorithm for supervised classification to identify the distribution of saline-alkaline land in the study area. The formula is as follows:

$$\min_{w, b, \xi_i} \left( \frac{1}{2} \|w\|^2 + C \sum_{i=1}^{n} \xi_i \right) \tag{1}$$

$$y_i (w \cdot x_i) \geq 1 - \xi_i, \xi_i \geq 0, i = 1, ..., n \tag{2}$$

In the formula, w represents the weight vector, which defines the direction of the hyperplane; b is the bias term, defining the offset of the hyperplane; $\xi_i$ is the slack

variable, which increases the robustness of the model; C is the regularization parameter, balancing the model complexity and training error; $y_i$     is the label of data point i, commonly used to define a hyperplane.

• The sentence on lines 169 to 174 "In this process, the slope data, image texture features....". It is a very long sentence, and the meaning is not clear, it should be rewritten in a clearer way.

**Respond:**Thank you for your suggestion. We have reorganized this part as follows:

Using the Normalized Difference Salinity Index (NDSI), slope data, and texture features as references, we employ a Support Vector Machine (SVM) algorithm for supervised classification to identify the distribution of saline-alkaline land in the study area.

• Line 172, the concept of field sample points appears for the first time, but it is not clear whether they refer to those taken on satellite images or are field work. The location and number of these sample points should also be clarified.

**Respond:**The description in this section was not clear enough. The sampling points here refer to the interpretation markers we determined on various land categories. We have rewritten this part as follows:

Based on the natural attributes of the soil in the study area, auxiliary data, and field survey conditions, we use high-resolution images from Google Maps to select interpretation markers for mild, moderate, and severe saline-alkaline land and other land types.

• Line 180 The criteria used to define low, moderate and severe salinity ranges should be explained in detail in the method section. Given that the results of the article show almost exclusively the area classified in each severity range, it is considered of vital importance to explain how these areas have been obtained and how the   accuracy of these results has been assessed.

**Respond:**Thank you for your suggestion. Initially, we established interpretation marker points on slight, moderate, and severe saline-alkaline lands, as well as other land categories, based on the natural attributes of the soil in the study area, auxiliary data, and field surveys. These marker points were then input into the remote sensing

images, and a supervised classification of the study area was conducted using the Support Vector Machine (SVM) algorithm. Finally, the accuracy of the supervised classification results was assessed using the confusion matrix method, which included overall classification accuracy and kappa coefficients. According to the accuracy results, the classification accuracies for the years 2002, 2007, 2012, 2017, and 2022 were 89.40%, 88.37%, 89.80%, 99.52%, and 96.83%, respectively, with kappa coefficients of 0.82, 0.81, 0.82, 0.99, and 0.95.

Regarding the results and discussion sections, it seems that the division of the two sections is not clear, especially in the discussion section, as this section shows new results and new figures and does not compare the results obtained with other previous studies. In fact, the discussion section is a continuation of the results, in the entire discussion section only two papers are cited, (Thorslund et al., 2021 and Jägermeyr et al., 2017), which is a clear sign that it is not a discussion per se, but a continuation of the presentation of the results, and what is more, conclusions are observed that are not justified by the results presented.

**Respond:** In the results section, we demonstrate the spatiotemporal changes of salinization in the Shiyang River Basin. The discussion section, on the other hand, focuses more on relating the salinization results to water conservancy projects, irrigation, reservoirs, etc. Based on your suggestion, we have revised the discussion section and added references.

• The analysis in the Temporal changes in soil salinization section should be rewritten to describe what is observed in the graphs, and it is useful to indicate the figure a, b, c, etc., to which you refer in each case.

**Respond:** We have rewritten the content of this section as follows:

The study area is divided into three parts based on natural landform units. The Southern Qilian mountain area did not exhibit soil salinization, so the focus is on the temporal changes in soil salinization area in the central corridor plains and the northern low hills and deserts (Fig 4). Over the 21 years, the change in the area of soil salinization in the basin was not substantial. Specifically, the area decreased from 2002 to 2007, increased from 2007 to 2012, and decreased again from 2017 to 2022.

In the central corridor plains, although the soil salinization area reached a historic low in 2007, overall, it shows an increasing trend. Compared to 2002, the area of soil salinization in 2022 increased by over 18%. The mild salinization area reached its maximum in 2017 and returned to the 2002 level by 2022. The increase in moderate salinization area is the most notable, while the area of severe salinization changed little. In the northern low mountains, hills, and desert areas, the soil salinization area showed an overall decreasing trend, with an average annual reduction rate of 38.19 km². The area of mild salinization changed little, with a slight increase in 2022 compared to 2002. The area of moderate salinization consistently decreased, especially significantly from 2002 to 2007. The area of severe salinization slowly decreased from 2002 to 2017 but increased from 2017 to 2022. Overall, the reduction in the soil salinization area in the northern low mountains and desert areas was significant, although the increase in the central corridor plains' soil salinization area was slightly less than the decrease in the northern areas.

• Lines 216 – 217: "From 2002 to 2022, the overall salinized area of the basin shows an increasing trend, with an average annual growth rate of 1881.9hm2/a". However, in fig 4a, the grey bar does not so clearly present this growth, in fact, from 2017 to 2022 there is a decrease in overall terms.

**Respond:** We have carefully reviewed the original text and data, and based on your suggestions, we have organized the salinization results according to the three natural divisions. From the time scale of 2002 to 2022, the overall change in salinization area is not significant. This is mainly due to the reduction in moderately salinized areas in the northern low hills and desert zones, while the moderately salinized areas in the central corridor plains have increased. Additionally, we have rewritten this part in the original text as follows:

Over the 21 years, the change in the area of soil salinization in the basin was not substantial. Specifically, the area decreased from 2002 to 2007, increased from 2007 to 2012, and decreased again from 2017 to 2022. In the central corridor plains, although the soil salinization area reached a historic low in 2007, overall, it shows an increasing trend. Compared to 2002, the area of soil salinization in 2022 increased by

over 18%. The mild salinization area reached its maximum in 2017 and returned to the 2002 level by 2022. The increase in moderate salinization area is the most notable, while the area of severe salinization changed little. In the northern low mountains, hills, and desert areas, the soil salinization area showed an overall decreasing trend, with an average annual reduction rate of 38.19 km². The area of mild salinization changed little, with a slight increase in 2022 compared to 2002. The area of moderate salinization consistently decreased, especially significantly from 2002 to 2007. The area of severe salinization slowly decreased from 2002 to 2017 but increased from 2017 to 2022. Overall, the reduction in the soil salinization area in the northern low mountains and desert areas was significant, although the increase in the central corridor plains' soil salinization area was slightly less than the decrease in the northern areas.

• Lines 214 and 218. The results are referred to as heavy salinization, the same nomenclature "severe" should be used.

**Respond:**We have made the necessary amendments in the original text.

• Lines 217 to 219:"shows an increasing trend", however it is maintained over time until the increase in the last period.

**Respond:**We have re-examined the data and found that the area of severe salinization in the basin remained relatively stable from 2002 to 2017, with little change. However, the area expanded from 2017 to 2022.

• Trends are analyzed by administrative counties, but the characteristics of each of them are not described: e.g. soil types, cultivated area, natural areas, semi-desertic areas, etc. Why is it analyzed by administrative terms and not for example by cluster of territory with similar characteristics? What are the factors that determine the response of the soil to the increase or decrease in salinity?

**Respond:**We also considered that it might be better to conduct the analysis based on natural landform divisions. Therefore, we have divided the study area into the Southern Qilian Mountain region, the Central Corridor Plains, and the Northern Low Hills and Desert region. There is no salinized soil in the Southern Qilian Mountain region, so we focused our analysis on the other two regions. In the inland river basins

of arid zones, factors such as irrigation, precipitation, and evaporation all influence soil salinity.

[Figure]

The description of the results in the irrigation districts, line 331 onwards, should be done graphically, and not with the names of the 27 districts because it is very confusing.

**Respond:**In Figure 7, we have labeled the location and degree of salinization for each irrigation area, and described it in the text.

• line 257, What do they mean when they talk about transfer of water for irrigation and the rise in the water table caused by the foreign water? Figure 5 seems key in the study but appears for the first time at the end.

**Respond:**Given the arid climate, low rainfall, and water scarcity in the Shiyang River Basin located in China's northwest arid region, a large number of water conservancy projects have been constructed to meet agricultural production needs through water transfer irrigation. The introduction of external water for irrigation leads to significant changes primarily in two areas: evaporation and groundwater levels. Firstly, as the amount of irrigation water increases, the soil moisture content rises, which under the unique conditions of arid regions, intensifies soil evaporation and plant transpiration, resulting in relatively low irrigation water use efficiency. Secondly, while external

water irrigation can temporarily alleviate the water resource scarcity in the basin, irrational irrigation practices may increase the risk of salinization and lead to further salt accumulation.Figure 5 is a conceptual diagram used in section 4.1 of the discussion, primarily to describe the salinization cycle in the study area. We have included annotations in the article.

• There is no relationship between conclusion (3) and the results shown and actually obtained in this document: the regional salinization problem is more prominent as a result of the rise in the groundwater level around the reservoirs, the evaporation from the irrigation of the agricultural fields, and the evaporation from the downstream ecological water conveyance. It should be completed and clarified. What data and results analyzed in this work support this conclusion?

Respond:Translation:Thank you for your careful review and feedback. Conclusion (3) is a summary of the discussion section. In section 4.1, we have demonstrated the salinization issues around the reservoirs. As the reservoir capacity increases, the surrounding soil salinization gradually worsens. Farmland irrigation primarily occurs in the central corridor plains, where saline-alkali land is widespread (Figure 7 labels the salinization degree of the irrigation areas). The annual precipitation is 150-300mm, whereas evaporation is as high as 1300-2000mm. The flood irrigation method and intense evaporation lead to salt accumulation in the surface soil, gradually worsening soil salinization. Downstream ecological water delivery primarily refers to water transfer projects to the Minqin area, where evaporation ranges from 2000-3000mm, and in some areas, precipitation is less than 150mm. Previous studies by our team have already indicated that in arid region farmland irrigation areas, non-productive water loss due to rainfall and irrigation reaches a peak proportion of 58%, and crop transpiration capabilities become a significant factor influencing evaporation loss. We have added references to some of our past work in the text.

Jiao Y, Zhu G, Meng G, et al. Estimating non-productive water loss in irrigated farmland in arid oasis regions: Based on stable isotope data[J]. Agricultural Water Management, 2023, 289: 108515.

• Line 370, in the statement: The relationship between salinity and irrigation is

apparent: over-irrigation increases the concentration of salts in the soil, leading to salinity problems. It should be clarified and related to which of the irrigation areas studied have over-irrigation and relate this aspect to the specific results obtained for these areas, in order to verify if what is stated is really what is happening in these areas.

**Respond:**In the Shiyang River Basin, the oasis irrigation area is the primary region for agricultural activities and is also where soil salinization is relatively concentrated. Due to the arid climate and low rainfall, irrigation is a crucial means of sustaining agricultural development. The irrigation method predominantly used in this area is flood irrigation, which has a significant impact on salinization. Upon correlating the salinization in the basin with the irrigation areas, we found that every irrigation area in the basin experiences soil salinization. Furthermore, the severity of soil salinization increases progressively from the midstream to the downstream regions. Our team has an observation station in the Shiyang River Basin (Northwest Normal University Shiyang River Basin Observation Station), and the farmland observation system has been continuously collecting data on irrigation practices, irrigation water volume, and crop planting structures for nearly 10 years.

As mentioned above, there are conclusions that are not justified by the results presented:

• Lines 381- 383, The following is stated: Farmland, grassland, and wasteland are at the most significant risk of being converted into saline soils, challenging farmland management. However, at no point in the article are the different types of crops analyzed by irrigation areas nor are results grouped into these categories of crop types provided.

**Respond:**Regarding the concern you mentioned, in our preliminary work, we processed land use type data in the Shiyang River Basin and calculated the land use transition matrix from 2002 to 2022. In the statistics of land type changes, saline-alkali land converted from farmland accounts for 3.07% of the saline-alkali land area in 2022, that converted from grassland accounts for 6.23%, and the largest proportion comes from barren land, which accounts for 11.27%. Therefore, we

presented this perspective in the conclusion section. Currently, we have added the corresponding calculation results in section 4.2 of the original text, as shown below:

The development of irrigated agriculture is necessary to meet the growing food needs of the global population (Jägermeyr et al., 2017). In the Shiyang River Basin, the saline-alkali land converted from farmland accounted for 3.07% of the saline-alkali land area in 2022. The conversion from grassland accounted for 6.23%, and the wasteland conversion rate was the highest, reaching 11.27%. These data indicate a significant risk of salinization occurring in farmland, grassland, and wasteland.

• The conclusions presented in lines 384 to 394 are not supported by the information and results presented in this work.

**Respond:**Firstly, external water transfer primarily refers to the water diversion projects in the Shiyang River Basin. In section 4.1, we mentioned that the Xiying River water diversion project began transferring water to Minqin in 2006, and the Jingdian Phase II project started diverting Yellow River water to the Minqin area in 2011. These water projects have significantly alleviated local water resource issues, and the salinized area in Minqin has been decreasing annually. However, we have also observed that the area of severe salinization is increasing. Therefore, while external water transfer can mitigate salinization to some extent, its impact on future water resources requires further study. Secondly, soil salinization around reservoirs is quite severe. Taking the Hongyashan Reservoir as an example, as the reservoir was constructed and expanded, soil salinization in the surrounding farmland worsened. Additionally, every irrigation area in the Shiyang River Basin has experienced soil salinization, which is widespread. There is a significant risk of farmland, grassland, and barren land being converted into saline soils. Thus, we propose that rational irrigation methods and land management are necessary to effectively address soil salinization issues in the future.

**Technical corrections:**

In figure 1:

I would recommend adding a more general situation map of the country,

adapting the legends of the rivers, and clarifying why the lines of the channels are cut; it seems that there is no connection to the hydrological network.

**Respond:** Thank you for your thoughtful reminder. We have added the boundary of the Eurasian continent in Figure 1 and marked the location of the Shiyang River Basin. The basin is located in the arid region of northwestern China, with very little rainfall and intense evaporation. These factors severely affect the basin's water resources, resulting in river discontinuities. Additionally, rivers in arid regions often have low flow volumes and can completely convert into groundwater when passing through loose surface deposits, later transforming back into surface water to form runoff in downstream areas. The hydrological network formed by these rivers and the constructed water conservancy projects are of great significance for agricultural irrigation in this inland river basin of the arid region.

Fig 1b is within the desert zone, however they look like crop areas.

**Respond:** Thank you very much for raising this question. We have modified the original Fig 1b (now Fig 1c). The revised figure1 are as follows:

[Figure]

Is Fig 1c in 3D? It is not clear what is being represented

**Respond:** The original Fig 1c (now Fig 1d) is not a 3D image, but rather a remote sensing image as the base map, on which we have overlaid the Hongyashan Reservoir

and some river elements. This approach allows for a more intuitive display of the spatial characteristics of this area. Due to the unique natural conditions and climate characteristics of the Shiyang River Basin, numerous water conservancy projects have been constructed throughout the basin to ensure agricultural development. These projects are mainly concentrated in the middle reaches of the basin, with the construction of the Hongyashan Reservoir having a particularly significant impact on ecological water supply to the downstream Minqin area. Additionally, the salinization issue around the reservoir is also a key focus of our research. For these reasons, we have specifically added the original Fig 1c (now Fig 1d).

In figure 3:

• In the figure, unnamed counties appear; if they are part of the studied basin area, they should be named. Clarify if they are ignored because the results do not show salinity in these areas.

**Respond:**In the original Figure 3, the unnamed counties are primarily located in the Southern Qilian mountain region of the basin, where soil salinization does not occur. Therefore, no analysis was conducted for these areas.

• It is difficult to follow the presentation of the results showed in this figure in the body of the text.

**Respond:**We have recreated Figure 3 and conducted descriptive analysis according to the natural landforms, as detailed below:

[Figure]

• The temporal evolution of salinity did not look very good to the naked eye in figures 3a to 3e. In addition to the figures presented, a single figure that represents, for each pixel, the relative temporal evolution throughout the time series (increase or decrease) would help in presenting the results.

Respond:Figure 3 primarily illustrates the spatial variation of soil salinization in the basin. To better observe this spatial variation, the new Figure 3 separately presents the salinization results for the central corridor plains and the northern low mountains, hills, and desert areas.

In figure 4, taking into account that results are represented with a spatial component, it is recommended to represent these results on a map with their spatial distribution.

Respond:We have reworked Figure 4 and redescribed it in the original text, as follows:

[Figure]

The Figure 5 is not cited in the text.

**Respond:**Thank you for your reminder. We have added the citations in the original text as shown below:

Ecological water transfer mainly refers to the Jingdian water transfer (i.e., water diversion from the Yellow River), which regulates the irrigation water use pattern by transferring water to Qingtuhu Lake through the Hongya Mountain Reservoir (Fig. 5).

In Figure 6, the results shown are not located correctly in space, it would be helpful to superimpose the scale, north, etc. on the aerial image.

**Respond:**Thank you for your suggestion, we have modified Figure 6, as shown below:

[Figure]

Figure 7, in this figure it seems that there is continuity of the channels and irrigation areas. Is that why they appear cut off in Fig. 1? This figure can be greatly improved, for example with maps of crop types in each area, soil types, etc., with data that provides information on the context of the space studied.

**Respond:**Regarding the issue of river disruption in Figure 1, this is primarily influenced by the topography and climate of the basin. The irrigation areas are mainly located where there are rivers, which is conducive to agricultural development. We have also optimized Figure 7 by adding crop types, as shown below:

[Figure]

Other minor comments:

• It is detected that a space is missing after the period and it often happens frequently throughout the document (e.g. 136, 142, 150, etc.)

**Respond:**Thank you for your careful pointing out. We have checked and revised the full text.

• line 144, It seems that the phrase "Detect more subtleties" is repetitive of the previous one.

**Respond:**Thank you for your careful pointing out. We have revised it in the original text.

• line 217, in units hm²/a What is the meaning of a? annual?

**Respond:** "/a" typically stands for "per annum," which means the average annual change.It is used to indicate the change per year.

---

## Referee Report (RR1)

**Journal:** Hydrology and Earth System Sciences

**Title:** Soil salinity patterns reveal changes in the water cycle of inland river basins in arid zones

**Authors:** Gaojia Meng, Guofeng Zhu, Yinying Jiao, Dongdong Qiu, Yuhao Wang, Siyu Lu, Rui Li, Jiawei Liu, Longhu Chen, Qinqin Wang, Enwei Huang, and Wentong Li

https://doi.org/10.5194/hess-2024-76

**General comments:**
The study has been significantly improved and has become clearer and more understandable. Unfortunately, I have been found still several shortcomings that need to be improved. Therefore, I recommend the manuscript for further minor revisions.

**Specific comments:**

Materials and Methods:

*Lines 104 – 108: General description of climatic conditions at the sites, include specific ranges of long-term meteorological variables (air temperatures, precipitation) for the period of the last 30 years (1991 – 2020).*

Thank you for adding the precipitation and evaporation rates of the location. Please, also add annual air temperature and main climate zone according to the Köppen–Geiger climate classification (1991 – 2020) the area is located. Do not forget to include a citation with a link to the list of references to the used climate classification.

*Lines 111 – 116: Change the soil classification to one of the international classification systems, e.g. "World reference base for soil resources 4th edition (2022)".*

Please add also include:

- soil texture for the specified soil unit,

- citation in the text of the classification system for the specified soils, for example: .....Cryosols, Leptosols, and Phaeozems (WRB, 2022). Include the classification system in the list of used literature.

**Thank you that I could oppose the manuscript of your article and the fundamental modifications of the Results and Discussion chapters. At the same time, I thank you for the constructive discussion on this issue. Good luck in your scientific career.**

---

## Referee Report (RR2)

Soil salinization in inland river basins of arid zones, driven by improper water resource use, significantly impacts agriculture and ecology. This study examines soil salinity changes in the Shiyang River Basin (2002–2022) using remote sensing and observational data, focusing on the effects of water conservancy projects, irrigation, and climate change. Key findings include: (1) a general increase in salinized areas and worsening salinization, (2) severe salinization in the lower reaches compared to the middle and upper reaches, and (3) human activities, such as rising groundwater levels near reservoirs, agricultural irrigation, and downstream water conveyance, as major contributors to salinization. Effective water resource management holds significant potential to mitigate soil salinization.

While these findings offer valuable insights for future research on soil salinity and its potential links to anthropogenic activities, the conclusions drawn in the study appear insufficiently justified. The analysis would benefit from quantitative work to more robustly support the claims about the role of human activities in salinization. Therefore, I suggest a moderate revision, incorporating a statistical approach to further strengthen the connection between human activities and the observed salinization trends before final conclusions are made. This would enhance the study's credibility and suitability for publication in the journal.

General comment:

One of the key conclusions of the study is that human activities have become a decisive factor in altering the salinization patterns of inland river basins. While this finding is significant, the evidence presented—namely, a simple comparison between irrigation areas and the regional distribution of soil salinization—does not provide sufficient support for such a conclusion. A more rigorous statistical approach is necessary to quantitatively assess the impact of human activities on salinization. For example, a time series analysis comparing the number of water conservancy projects constructed in the basin over the last decade with trends in soil salinization could offer stronger evidence. Similarly, comparing irrigation levels with salinization trends in a statistical manner would help substantiate the argument that increasing salinization is primarily driven by human activities, rather than being solely attributed to natural climate changes over the past decade. This more thorough analysis would significantly strengthen the study's conclusions.

Specific comments:

**1**
The "Background Conditions of the Study Area" section would benefit from additional climate information. Providing more detailed climate data in a numeric way, such as temperature, precipitation patterns, and seasonal variations, would offer a clearer understanding of the environmental context of the study area.

**2**
For better clarity, I recommend consolidating all data sources into a single comprehensive table. This will provide a more transparent overview of the data leveraged in the study and allow readers to easily assess the different datasets used.

**3**

In the discussion section, it would be helpful to include a more thorough examination of the potential limitations of the study. Discussing factors such as data constraints, assumptions, or other uncertainties would strengthen the study's credibility and provide a balanced perspective.

**4**

Please specify the country in which the study sites are located. This information is essential for providing geographical context to the research and enhancing its clarity for readers.

**5**

In the discussion section, please clarify or justify whether the study region can be considered representative of typical inland river basins. This will help contextualize the findings.

---

## Referee Report (RR3)

Manuscript Number:HESS-2024-76 , "Soil salinity patterns reveal changes in the water cycle of inland river basins in arid zones" by Meng et al.

**General comments:**

The manuscript has been reviewed by two experts in the previous round, and the author substantially revised the manuscript based on the comments from reviewers and editor. I think the manuscript has been improved in quality and logic after the revision. I only has some minor comment on this manuscript.Overall, the authors have invested considerable effort in writing and revising this article. I recommend that the article be accepted with minor revisions.

Minors:

(1) Lines 251-256: The subplot labels in Figure 5 are not clear enough, and the legend position needs adjustment.

(2) Line 281: The font size of the y-axis title in Figure 6 is too small.

(3) Line 407: The irrigation district numbers in Figure 10 are too small and difficult to identify.

(4) Line 101: "Xiyang River" should be "Xiying River".

(5) Line 241: "especially in the downstream" should be "especially in the downstream area".

(6) Line 298: "altered evaporation process" is not accurately expressed.

(7) Line 444: "small variations" is not precise enough in wording.

(8) Line 490: Abu Hammad citation format is incorrect, should be "Abu Hammad and Tumeizi".

(9) Verify the references in the text. Since papers on the relationship between saline-alkali land and hydrology are relatively limited, although such research is very important, it is recommended to add some books or research reports as references. For example:

Saline-alkali Soil Science and Comprehensive Utilization (Hu et al., 2025); Practical Q&A

and Case Analysis of Saline-alkali Land Improvement Technology (Liang et al., 2018).

---

## Author Response (AR2)

**Response to Editor:**

1. I did not think the authors take paper publication as a serious academic activity. All figures need good quality, and proper font size to be recognized. Also the references should fit HESS's format.

**Respond:** Thank you for your suggestions. We have checked all figures, ensuring that their resolution is above 600 dpi and that the information is identifiable. At the same time, we have modified the reference format to ensure compliance with HESS requirements, and the revised content has been added to the new manuscript and marked in blue.

2. The serious problem of current paper is the lack of in-situ validation. The authors claim that "the severity of salinization has been increasing", however, I did not see the validation of the remote sensing results. Without validation, it does not warrant a scientific paper.

**Respond:**

**(1) Cross-validation of remote sensing data**

[revised manuscript text omitted]

Figure R3. Experimental Field

3. From Figure 1 and 3, we cannot clearly see the salinity changes. I suggest the authors zoom in the satellite images, and compare for example two images before and after salinity, and highlight the salinitilized area. This can further strengthen your conclusion on the increasing of salinity in this region.

**Respond:** In the revised manuscript, Figure 1 is an overview of the study area, where Figures a and b represent two main aspects of the research, representing salinization caused by farmland irrigation and ecological water transfer, respectively. In Figure a, we selected the downstream Qingthu Lake, which shows significant salt distribution characteristics around it. Figure b focuses on salinization phenomena in oasis irrigation farmlands. Figure 3 is a schematic diagram showing the spatial distribution of salinization processes in the Shiyang River Basin. Considering the large basin area and dispersed salt distribution, we divided it into the middle and downstream sections to demonstrate its changes, with orange circles marking areas of significant variation. As shown below:

[Figure]

**Figure 1.** Overview map of the study area (a: Location distribution map of the Shiyang River Basin;

b: Qingtu Lake (from USGS); c: Saline soils in agricultural land (from Google Maps); d:

Distribution of water systems in the Shiyang River Basin (from USGS))

[Figure]

**Figure 5.** Spatial Distribution Map of Salinization in the Shiyang River Basin (a: Distribution of soil salinization in the northern hills and oasis-desert transition zone in 2002; b-c: Expansion areas in soil salinization in the northern hills and oasis-desert transition zone in 2012 and 2022; d: Distribution of soil salinization in the central corridor plain in 2002; e-f: Expansion areas in soil salinization in the central corridor plain in 2012 and 2022)

4. I suggest the authors separate apart and map out different mechanisms to explain salinity changes. For example, I am curious to see where the irrigation plays a dominant role in salinity, and where reservoir inundation area is a key factor. Using two conceptual diagrams to demonstrate these two mechanisms is likely a good idea.

I'm curious to know: is vegetation a good indicator for salinity? Can you do some analysis to test it? Does salinity have significant impacts on vegetation's root zone (https://hess.copernicus.org/articles/28/4477/2024/)?

**Respond:** We have already created two conceptual diagrams in the discussion section:

one illustrating the water conservancy project and salt cycle, and another depicting the water and salt cycle in agricultural irrigation, and have accordingly modified the discussion section, as stated in the following red text. Moreover, existing literature has proven that salinity significantly impacts vegetation, with high-salinity environments causing restricted root growth, reduced photosynthetic efficiency, and decreased biomass (Perri et al., 2020). Therefore, vegetation distribution characteristics can theoretically serve as an indirect indicator of salinity. However, our experimental research discovered clear limitations in using vegetation coverage as a single indicator. Different plants have varying salt tolerance, and focusing solely on vegetation coverage would overlook critical physiological and morphological changes in vegetation. Given the complexity of the research system, we plan to select experimental fields and forest-grassland sample plots in subsequent research for more cautious and systematic experimental validation, to reveal the intricate mechanisms between salinity and vegetation.

[Figure]

**Figure 7.** The process of salinization caused by reservoirs.

[Figure]

Figure 9.The process of salinization caused by agricultural irrigation

I cannot follow some statements:

Line 40-41: water bodies also impact soil quality, mainly through irrigation and precipitation. How water bodies can impact soil quality through precipitation?

**Respond:** The wording in this part was indeed ambiguous. The original intention was to express that water bodies, through two methods of precipitation and irrigation, can change groundwater levels and salt distribution, thereby affecting the physical and chemical properties of the soil. We have already reviewed and modified the entire text.As shown below:

Soil plays a critical role in promoting plant growth, regulating precipitation infiltration and distribution to coordinate watershed water cycles. Moreover, its purification capacity enables the decomposition of potential pollutants, thereby preventing water and air pollution to a certain extent (Bünemann et al., 2018; Renshu et al., 2024). However, once soil quality declines or undergoes degradation, it can cause irreversible damage and directly impact human life (Reynolds et al., 2007; Abu Hammad and Tumeizi, 2012).

There are many such confusing statements. Might be a good idea to invite an English native speaker, to carefully improve the writing all through this MS. My gut initial decision was rejection due to the still poor quality after giving a chance to

revision. However, due to the importance of salinity issue in this region, I still don't want to kill this paper. I give the authors one more and LAST chance to revise it. Hope to receive your good quality paper soon. If you need more time to do it, please let me know. I can extend the deadline.

**Respond:** Thank you for giving us the opportunity to revise our article and for providing valuable revision suggestions. We greatly appreciate this opportunity to revise, and we have thoroughly revised the entire manuscript according to your and the reviewers' recommendations. Thank you again.

**Response to Reviewer#1:**

Thank you for your valuable feedback, which has greatly improved our manuscript. I deeply appreciate your constructive suggestions and will carefully address each one with detailed responses.

In the revised manuscript, we have revised the content of the manuscript. **The revisions in the manuscript are indicated using blue font.** Below is a comprehensive overview of the modifications we have made:

**General comments:**

The study has been significantly improved and has become clearer and more understandable. Unfortunately, I have been found still several shortcomings that need to be improved. Therefore, I recommend the manuscript for further minor revisions.

**Respond:**Thank you once again for your valuable comments on the manuscript. They are very important for further improving the quality of the article. We have carefully revised the issues you raised.

**Specific comments:**

Materials and Methods:

1. Lines 104 – 108: General description of climatic conditions at the sites, include specific ranges of long-term meteorological variables (air temperatures, precipitation) for the period of the last 30 years (1991 – 2020).

Thank you for adding the precipitation and evaporation rates of the location. Please, also add annual air temperature and main climate zone according to the Köppen‑Geiger climate classification (1991-2020) the area is located. Do not forget to include a citation with a link to the list of references to the used climate classification.

**Respond:** Thank you for your suggestions. We have added the annual mean temperature to the manuscript and identified the climate zones of the study area based on the Köppen‑Geiger climate classification (1991‑2020), along with the corresponding references, as shown below:

The study area is located in the BWK climate zone under the Köppen-Geiger climate classification, which is a cold arid desert climate (Beck et al., 2018; Beck et al., 2023). It features strong solar radiation, intense evaporation, significant diurnal temperature variation, and an annual average temperature below 8°C. Precipitation is sparse and primarily influenced by westerlies and monsoons. Mountain areas receive more precipitation than plains, with higher precipitation during summer and autumn, and significantly less during winter and spring. The terrain slopes from southwest to northeast and is divided into three units.

Please add also include:

- soil texture for the specified soil unit,

- citation in the text of the classification system for the specified soils, for example: .....Cryosols, Leptosols, and Phaeozems (WRB, 2022). Include the classification system in the list of used literature.

**Respond:**Thank you for your suggested revisions. We have added information on the soil texture of the soil units and included references for the soil classification system, as shown below:

The bedrock of the southern Qilian Mountains consists of metamorphosed sandstones and volcanic rocks, with soil textures predominantly coarse and medium,

including Cryosols, Leptosols, and Phaeozems (WRB, 2022). The land is primarily forest and grassland, with annual precipitation of 300-600mm, evaporation rates of 700-1200mm, and the groundwater level is 50-200 meters below the surface. The central corridor plain features bedrock composed of schist and slate, with soil textures predominantly medium and fine, including Gypsisols, Calcisols, and Solonchaks. The land use is primarily agricultural, with annual precipitation of 150-300mm, evaporation rates of 1300-2000mm, and the groundwater level is 15-50 meters below the surface. The bedrock of the northern hills and oasis-desert transition zone is predominantly igneous rock, with soil textures mainly coarse, including Arenosols, Leptosols, and Solonchaks. The landscape is barren, with annual precipitation below 150mm, evaporation rates of 2000-3000mm, and the groundwater level is 10-30 meters below the surface. The three geomorphological units show distinct differences, with increasing aridity from south to north.

References:

IUSS Working Group WRB: World Reference Base for Soil Resources. International soil classification system for naming soils and creating legends for soil maps, 4th edition, International Union of Soil Sciences (IUSS), Vienna, Austria, ISBN 979-8-9862451-1-9, 2022.

**Response to Reviewer#2:**

Your valuable insights have significantly contributed to enhancing the quality of our manuscript. I feel extremely honored to receive such positive and constructive feedback from you. I genuinely appreciate every thoughtful suggestion you've provided and will address each one of them with utmost care, offering detailed responses in return.

In the revised manuscript, we have meticulously restructured and refined the logic and content of the abstract, introduction, discussion, and image sections. **The revisions in the manuscript are indicated using blue font.** Below is a comprehensive overview of the modifications we have made:

**General comments:**

One of the key conclusions of the study is that human activities have become a decisive factor in altering the salinization patterns of inland river basins. While this finding is significant, the evidence presented—namely, a simple comparison between irrigation areas and the regional distribution of soil salinization—does not provide sufficient support for such a conclusion. A more rigorous statistical approach is necessary to quantitatively assess the impact of human activities on salinization. For example, a time series analysis comparing the number of water conservancy projects constructed in the basin over the last decade with trends in soil salinization could offer stronger evidence. Similarly, comparing irrigation levels with salinization trends in a statistical manner would help substantiate the argument that increasing salinization is primarily driven by human activities, rather than being solely attributed to natural climate changes over the past decade. This more thorough analysis would significantly strengthen the study's conclusions.

**Respond:** Thank you very much for your insights. To provide more comprehensive content, we have diligently reviewed and refined the abstract, introduction, literature review, research methods,results, discussion and conclusion, aiming to enhance the depth of our research focus.

**Specific comments:**

**1**

The "Background Conditions of the Study Area" section would benefit from additional climate information. Providing more detailed climate data in a numeric way, such as temperature, precipitation patterns, and seasonal variations, would offer a clearer understanding of the environmental context of the study area.

**Respond:** Thank you for your suggestions. We have added more detailed climate data, including climate classification, annual average temperature, and precipitation patterns, in the "Background Conditions of the Study Area" section, as shown below:

The Shiyang River Basin is located in northwestern China, at the eastern end of the Hexi Corridor. It consists of eight major tributaries: the Dajing River, the Gulang River, the Huangyang River, the Zaomu River, the Jinta River, and the Xiyang River (Fig. 1). Lakes and wetlands in the whole region mainly exist in reservoirs, with 15 reservoirs built with a more than 1 million cubic meters capacity. Water storage in reservoirs helps to adjust the distribution of river water. The study area is located in the BWK climate zone under the Köppen-Geiger climate classification, which is a cold arid desert climate (Beck et al., 2018; Beck et al., 2023). It features strong solar radiation, intense evaporation, significant diurnal temperature variation, and an annual average temperature below 8°C. Precipitation is sparse and primarily influenced by westerlies and monsoons. Mountain areas receive more precipitation than plains, with higher precipitation during summer and autumn, and significantly less during winter and spring. The terrain slopes from southwest to northeast and is divided into three units. The bedrock of the southern Qilian Mountains consists of metamorphosed sandstones and volcanic rocks, with soil textures predominantly coarse and medium, including Cryosols, Leptosols, and Phaeozems (WRB, 2022). The land is primarily forest and grassland, with annual precipitation of 300-600mm, evaporation rates of 700-1200mm, and the groundwater level is 50-200 meters below the surface. The central corridor plain features bedrock composed of schist and slate, with soil textures predominantly medium and fine, including Gypsisols, Calcisols, and Solonchaks. The land use is primarily agricultural, with annual precipitation of 150-300mm, evaporation rates of 1300-2000mm, and the groundwater level is 15-50 meters below

the surface. The bedrock of the northern hills and oasis-desert transition zone is predominantly igneous rock, with soil textures mainly coarse, including Arenosols, Leptosols, and Solonchaks. The landscape is barren, with annual precipitation below 150mm, evaporation rates of 2000-3000mm, and the groundwater level is 10-30 meters below the surface. The three geomorphological units show distinct differences, with increasing aridity from south to north.

**2**

For better clarity, I recommend consolidating all data sources into a single comprehensive table. This will provide a more transparent overview of the data leveraged in the study and allow readers to easily assess the different datasets used.

**Respond:** Thank you for your valuable suggestions. We have organized all the data sources in Table 1 to present the dataset more clearly, making it easier for readers to understand, as shown below:

**Table 1.** List of data products used in the study

| Products | Temporal resolution | Spatial resolution | Temporal coverage | Data Source |
|----------|---------------------|--------------------|--------------------|-------------|
| Landsat-5 | 16d | 30m | 1984-2013 | https://earthxplorer.usgs.gov |
| Landsat-7 | 16d | 30m | 1999-present | https://earthxplorer.usgs.gov |
| Landsat-8 | 16d | 30m | 2013-present | https://earthxplorer.usgs.gov |
| Landsat-9 | 16d | 30m | 2021-present | https://earthxplorer.usgs.gov |
| Landuse | Annual | 30m | 1985-2022 | https://zenodo.org/records/8176941 |
| ASTER GDEM | / | 30m | 2000-2019 | http://reverb.echo.nasa.gov/reverb/ |

**3**

In the discussion section, it would be helpful to include a more thorough examination of the potential limitations of the study. Discussing factors such as data constraints, assumptions, or other uncertainties would strengthen the study's credibility and provide a balanced perspective.

**Respond:** Thank you for your suggestion. Incorporating uncertainty analysis in the discussion is indeed very important. Therefore, we have included a discussion on the uncertainties related to the data and other aspects in the manuscript, as shown below:

This study analyzed soil salinization in the Shiyang River Basin using Landsat satellite data. However, due to the inherent uncertainties of satellite data, the results

may have certain limitations. Although satellites can provide multispectral data, the spectral resolution is relatively low, and atmospheric correction issues may also affect data accuracy, posing challenges for identifying soil salinization (Vicente-Serrano et al., 2008; Vanonckelen et al., 2013). Landsat has a revisit cycle of 16 days, which can be further extended by climatic effects during certain seasons, significantly limiting seasonal monitoring of the region. Additionally, the selection and quantity of training data directly affect the accuracy of supervised classification. An accuracy assessment of the supervised classification results revealed classification accuracies of 89.40%, 88.37%, 89.80%, 99.52%, and 96.83% for the years 2002, 2007, 2012, 2017, and 2022, respectively, with kappa coefficients of 0.82, 0.81, 0.82, 0.99, and 0.95. However, due to the limitations of sampling size and satellite data, the identification of mildly saline-alkaline land is slightly less effective compared to other types of land, which requires further improvement in future work. Because soil salinization is influenced by multiple interacting factors such as climate and irrigation, single-satellite data alone struggle to fully capture the variation of all environmental components. Future research will expand data sources by integrating field measurements, meteorological records, and irrigation information to obtain more comprehensive or higher-resolution multi-source fusion data. Our systematic soil salinity monitoring for this basin began in 2019, which represents a limited timeframe that prevents us from comprehensively validating remote sensing interpretation results using long-term soil physicochemical parameter data. Consequently, current accuracy assessments primarily relies on field-verified sample points collected between 2019 and 2024, which somewhat constrains our ability to verify the long-term dynamic processes of saline land changes. Nevertheless, these validation points still provide important ground truth references for remote sensing monitoring results. Moreover, the application of deep learning models for image classification and feature extraction could deepen our understanding of the driving mechanisms behind soil salinization distribution, thereby improving the applicability of such findings in hydrology and soil management.

**4**

Please specify the country in which the study sites are located. This information is essential for providing geographical context to the research and enhancing its clarity for readers.

**Respond:** Thank you for your suggestion. In the "The Background Conditions of the Study Area" section, we have already specified that the study area is located in the northwestern part of China, and Figure 1 indicates its location on the Eurasian continent, as shown below:

The Shiyang River Basin is located in northwestern China, at the eastern end of the Hexi Corridor. It consists of eight major tributaries: the Dajing River, the Gulang River, the Huangyang River, the Zaomu River, the Jinta River, and the Xiyang River (Fig. 1).

[Figure]

**Figure 1.**Overview map of the study area (a:Location distribution map of the Shiyang River Basin; b: Qingtu Lake (from USGS); c: Saline soils in agricultural land (from Google Maps); d: Distribution of water systems in the Shiyang River Basin (from USGS))

**5**

In the discussion section, please clarify or justify whether the study region can be considered representative of typical inland river basins. This will help contextualize

the findings.

**Respond:** Thank you for your valuable suggestion. Regarding whether the research area can be considered a representative of typical inland river basins, we have clarified this in the discussion section and cited relevant literature. We added a detailed analysis of the region's representativeness in the discussion section, as shown below:

4.1 Soil salinization and basin water conservancy project

[revised manuscript text omitted]

---

## Author Response (AR3)

**Response to Editor:**

While both reviewers are satisfied with the revisions overall, some minor revisions are still needed before acceptance. I look forward to your updated manuscript.

**Respond:** Thank you for your long-term attention and guidance to the thesis. These suggestions are of great value for improving the quality of the article, and we have completed the corresponding revisions based on the opinions. Thank you sincerely again.

**Response to Reviewer#4:**

Thank you for your comments. We take each suggestion very seriously and will provide a detailed response. In the revised manuscript, we have revised the content of the manuscript. **The revisions in the manuscript are indicated using blue font.** Below is a comprehensive overview of the modifications we have made:

**General comments:**

The manuscript has been reviewed by two experts in the previous round, and the author substantially revised the manuscript based on the comments from reviewers and editor. I think the manuscript has been improved in quality and logic after the revision. I only has some minor comment on this manuscript.Overall, the authors have invested considerable effort in writing and revising this article. I recommend that the article be accepted with minor revisions.

**Respond:** Thank you for your valuable comments on the article. We have checked and revised the paper according to your suggestions.

**Minors comments:**

(1) Lines 251-256: The subplot labels in Figure 5 are not clear enough, and the legend position needs adjustment.

**Respond:** We have revised the subplot labels in Figure 5 and optimized the legend, as shown below:

[Figure]

**Figure 5.** Spatial Distribution Map of Salinization in the Shiyang River Basin (a: Distribution of soil salinization in the northern hills and oasis-desert transition zone in 2002; b-c: Expansion areas in soil salinization in the northern hills and oasis-desert transition zone in 2012 and 2022; d: Distribution of soil salinization in the central corridor plain in 2002; e-f: Expansion areas in soil salinization in the central corridor plain in 2012 and 2022)

(2) Line 281: The font size of the y-axis title in Figure 6 is too small.

**Respond:** We have moderately adjusted the y-axis of Figure 6 to make it easier to observe, as shown below:

[Figure]

(3) Line 407: The irrigation district numbers in Figure 10 are too small and difficult to identify.

**Respond:** Thank you for your suggestions, and we have made the modifications, as shown below:

Surface water and groundwater irrigation are the primary irrigation methods in the Shiyang River Basin, significantly impacting soil salinization in both agricultural and non-agricultural areas (Fig.9). The Shiyang River Basin comprises 27 irrigation

districts (Fig.10), with seriously salinized districts concentrated in the middle and lower reaches, while non-salinized districts are located in the upstream region. Severe soil salinization includes Hongyashan Irrigation District (HYSID), Changning Irrigation District (CNID), and Huanhe Irrigation District (HHID). Moderate soil salinization includes Dongdahe Irrigation District (DDHID), Nanhu Irrigation District (NHID), Donghe Irrigation District (DHID), Xiyinghe Irrigation District (XYHID), Siba Irrigation District (SBID), and Qinghe Irrigation District (QHID). Among these, DDHID experienced particularly severe soil salinization, with a significant increase in salinized area during 2007–2012. Mild soil salinization includes Gulanghe Irrigation District (GLHID), Wujiaojing Irrigation District (WJJID), Huangyanghe Irrigation District (HYHID), Yinhuang Irrigation District (YHID), Qiduntai Irrigation District (QDTID), Jingdian Irrigation District (JDID), Dajinghe Irrigation District (DJHID), Qingyuanjing Irrigation District (QYJID), Zamuhe Irrigation District (ZMHID), Jintahe Irrigation District (JTHID), Jingtaichuan Irrigation District (JTCID), Jinyangjingyuan Irrigation District (JYJYID), Jinchuan Irrigation District (JCID), Yongchang Irrigation District (YCID), and Xihe Irrigation District (XHID). GLHID and WJJID showed a continuous increase in salinized area from 2002 to 2017 but experienced a reduction from 2017 to 2022, while other districts saw minimal changes in salinization area. Zhangyi Irrigation District (ZYID), Gufenghe Irrigation District (GFHID), and Tuiguangzhan Irrigation District (TGZID) have no salinization. Overall, irrigation is the main factor influencing the gradual increase in soil salinization from upstream to downstream in the Shiyang River Basin, highlighting the profound impact of human agricultural activities on salinization in the basin.

[Figure]

Figure 10. Distribution of irrigation areas in the Shiyang River Basin

(4) Line 101: "Xiyang River" should be "Xiying River".

**Respond:** Thank you for your suggestions. We have corrected the erroneous expressions in the original text, as shown below:

It consists of eight major tributaries: the Dajing River, the Gulang River, the Huangyang River, the Zamu River, the Jinta River, and the Xiying River (Fig. 1).

(5) Line 241: "especially in the downstream" should be "especially in the downstream area".

**Respond:** Thank you for raising the questions, and we have made the revisions, as shown below:

The results showed that the salinization of the basin gradually increased from upstream to downstream, especially in the downstream area of the basin near Qingtu Lake, where the salinization of the soil was the most serious.

(6) Line 298: "altered evaporation process" is not accurately expressed.

**Respond:** Thank you for raising the questions. Our original intention was to express that external water diversion irrigation directly increases surface and soil water

content by artificially introducing water sources, a process that disrupts the original "precipitation-evaporation-infiltration" balance. Meanwhile, we have revised the language in the original text, as shown below:

Its negative effects are reflected in two aspects: the evaporation process altered by the introduced water for irrigation, and the rise of groundwater level caused by the inflow of external water (Duan et al., 2022).

(7) Line 444: "small variations" is not precise enough in wording.

**Respond:** Thank you for pointing out the issues, and we have made the amendments, as shown below:

The basin's salinization area showed overall minimal variation, but salinity gradually intensified from southwest to northeast.

(8) Line 490: Abu Hammad citation format is incorrect, should be "Abu Hammad and Tumeizi".

**Respond:** We have checked and corrected the citations in the article, as shown below:

Abu Hammad, A. and Tumeizi, A.: Land degradation: socioeconomic and environmental causes and consequences in the eastern Mediterranean, Land Degrad. Dev., 23(3), 216-226, https://doi.org/10.1002/ldr.1069, 2012.

(9) Verify the references in the text. Since papers on the relationship between saline-alkali land and hydrology are relatively limited, although such research is very important, it is recommended to add some books or research reports as references. For example:Saline-alkali Soil Science and Comprehensive Utilization (Hu et al., 2025); Practical Q&A and Case Analysis of Saline-alkali Land Improvement Technology (Liang et al., 2018).

**Respond:** Thank you for your suggestions. We have added the relevant citations to the paper, as shown below:

Soil salinization can be classified into primary salinization and secondary salinization based on its causes (Liang et al., 2018).

Remote sensing technology has been widely used to assess soil salinization, and feature spectral characteristics are essential markers for identifying saline soils (Ivushkin et al., 2019; Hu et al., 2024).

Hu, S., Li, Rong., and Gao, H.: Saline-alkali Soil Science and Comprehensive Utilization, Science Press., ISBN 978-7-030-80064-0, 2024.

Liang, F., Li, Z., and Zhang, L.: Practical Q&A and Case Analysis of Saline-alkali Land Improvement Technology, China Agriculture Press ., ISBN 978-7-109-24618-8, 2018.